# Fine-grained, spatio-temporal datasets measuring 200 years of land development in the United States

Johannes H. Uhl[1,5], Stefan Leyk[1,5], Caitlin M. McShane[1], Anna E. Braswell[2], Dylan S. Connor[3], and Deborah Balk[4]

[1]Department of Geography, University of Colorado Boulder, Boulder, CO 80309, USA
[2]Earth Lab, Cooperative Institute for Research in Environmental Sciences (CIRES), University of Colorado Boulder, Boulder, CO 80303, USA
[3]School of Geographical Sciences & Urban Planning, Arizona State University, Tempe, AZ 85281, USA
[4]CUNY Institute for Demographic Research & Baruch College, City University of New York, New York City, NY 10017, USA
[5]Institute of Behavioral Science, University of Colorado Boulder, Boulder, CO 80309, USA

**Correspondence:** Johannes H. Uhl (johannes.uhl@colorado.edu)

**Abstract.** The collection, processing and analysis of remote sensing data since the early 1970s has rapidly improved our understanding of change on the Earth's surface. While satellite-based earth observation has proven to be of vast scientific value, these data are typically confined to recent decades of observation and often lack important thematic detail. Here, we advance in this arena by constructing new spatially-explicit settlement data for the United States that extend back to the early nineteenth
century, and is consistently enumerated at fine spatial and temporal granularity (i.e., 250 m spatial, and 5 a temporal resolution). We create these time series using a large, novel building stock database to extract and map retrospective, fine-grained spatial distributions of built-up properties in the conterminous United States from 1810 to 2015. From our data extraction, we analyse and publish a series of gridded geospatial datasets that enable novel retrospective historical analysis of the built environment at unprecedented spatial and temporal resolution. The datasets are available at https://dataverse.harvard.edu/dataverse/hisdacus
(Uhl and Leyk, 2020a, b, c, d).

## 1   Introduction

Over the last two hundred years, the number of people living in urban areas in the United States has grown more than 800-fold, from around 320 thousand, and 6% of the population, in 1800 to 270 million, and 80% of the population, by 2016 (US Census Bureau, 1993, 2016). The urbanization of the United States has produced vast metropolitan areas and an array of smaller to
mid-size settlements, reorganizing the population and land structure of the continent in the process. Despite being critical to understanding the changes and coupling mechanisms underlying human and natural systems, our knowledge of settlement and development in the United States (and elsewhere) is far from complete. Understanding these long-term changes is both of historical interest, and crucial for the reliable projection of future change. These are challenging issues to contend with, especially as, prior to the post-1970 era of remote-sensing based earth observation and digital cartography, there is a serious
scarcity of structured historical geospatial data.

In previous work, we presented the Historical Settlement Data Compilation for the U.S. (HISDAC-US), a novel database that enables analysis of fine-resolution settlement and urban development patterns at five-year intervals from 1810 to 2015 (Leyk and Uhl, 2018a). This long timeframe of observation is one of the distinguishing features of the HISDAC-US, which is providing unprecedented opportunities for studying long-term settlement and development trends. To date, the HISDAC-US contains two main gridded data products: (a) a built-up intensity surface series (BUI; Leyk and Uhl, 2018b), mapping the approximate building indoor area of all built-up structures within each 250x250 m grid cell in the conterminous U.S., and (b) a temporal composite surface, mapping the year when a grid cell was first built up, the "first built-up year" (FBUY; Leyk and Uhl, 2018c). The BUI surface series represents an aggregated, volumetric measure of built-up intensity, the total indoor floor area present within a fixed area. However, as noted in our previous work, these retrospective estimates of built-up intensity will be less accurate in areas that have undergone substantial building replacement or remodelling activities (Leyk and Uhl, 2018a).

Herein, we introduce two significant developments in the HISDAC-US that allow for more generic and unbiased analytical characterization of long-term building patterns in the United States. These new, gridded, spatial time-series data map a) counts of built-up property records, (i.e., representing individually owned buildings or building units) and b) counts of unique built-up property locations (i.e., physical structures, disregarding the ownership situation), at 250 m spatial resolution and for each half-decade (i.e., 5-year intervals) from 1810 to 2015. We derived these counts from vast numbers of cadastral records contained in the Zillow Transaction and Assessment Dataset (ZTRAX; Zillow Inc., 2016). These additions to the HISDAC-US provide an important step beyond our previously published BUI surfaces: They enable reconstruction of fine-grained historical building densities for much of the United States and have applications illustrated in various research efforts leveraging the HISDAC-US to study urban geography (Uhl et al., 2020), historical demography (Leyk et al., 2020), road network evolution (Boeing, 2020), population allocation (Leyk et al., 2019), natural hazards and extreme events (Balch et al., 2020; Iglesias and Travis, 2020; Mietkiewicz et al., 2020; Braswell et al.), landscape fragmentation (Millhouser, 2019), and popular science (Financial Times, 2020).

The generation of these new products has been driven by the ongoing "data revolution" (Kitchin, 2014), which has spurred rapid advancements in web-based data storage and distribution infrastructure, high-performance computing, and the expansion of public and private open-data policies. The decision by U.S. county-level administrations to publicly share rich cadastral and tax assessment data, and the acquisition and harmonization of these data by the real-estate company, Zillow Group Inc., has been particularly important for our work. Through their efforts, Zillow has produced ZTRAX, a large building stock and property database holding millions of records on built-up properties and their characteristics, including building size, land-use type, age, and property value. Zillow has recently made ZTRAX available for scientific research via institutional data share agreements and it has recently been employed by researchers in various scientific disciplines (e.g., Bernstein et al., 2019; Boslett and Hill, 2019; Clarke and Freedman, 2019; Gindelsky et al., 2019; Kim et al., 2019; Peng and Zhang, 2019; Tarafdar et al., 2019; Uhl et al., 2019; Zoraghein and Leyk, 2019; Baldauf et al., 2020; Bechard, 2020; Buchanan et al., 2020; Connor et al., 2020; D'Lima and Schultz, 2020; Nolte, 2020; Onda et al., 2020; Shen et al., 2020; Stern and Lester, 2020; Wentland et al., 2020). We have continued to leverage this novel and unique data source in producing and advancing the HISDAC-US.

HISDAC-US consists of a variety of gridded surface datasets (i.e., geospatial raster layers) measuring different characteristics of the built environment and provides an unprecedented data source for longitudinal geographic and demographic research. HISDAC-US exploits the "year built" attribute provided by ZTRAX, reporting the year when a built-up structure has been established. This attribute is derived from historical, county-level tax assessment data records and is available for more than 117 million built-up structures in the U.S. The detailed spatial and temporal information provided in ZTRAX allows for mapping retrospective distributions of human settlement and colonial land development at unprecedented spatial and temporal granularity (i.e., 250 m spatial and 5 a temporal resolution), and extends across an unmatched time period. Hence, these data help overcome several fundamental temporal and spatial limitations in data sources widely used by the Earth system science community such as the Global Human Settlement Layer (GHSL; Pesaresi et al., 2013), the World Settlement Footprint Evolution dataset (Marconcini et al., 2020), the National Land Cover Database (NLCD; Homer et al., 2007), the Gridded Rural-Urban Mapping Project (GRUMP; CIESIN, 2004) or multi-temporal population datasets (e.g., Gridded Population of the World (GPW; Balk and Yetman, 2004), WorldPop (Tatem, 2017), GHS-POP (Freire et al., 2000), or LandScan (Dobson et al., 2000) (see an overview in Leyk et al., 2019)[1], as well as sparse and more computationally expensive and labour-intensive alternatives such as historical and archaeological records (Reba et al., 2016; Hedefalk et al., 2017; Ostafin et al., 2020; Lieskovskỳ et al., 2018), georeferenced social science data (Kugler et al., 2019), data extracted from historical maps (Uhl et al., 2019; Kaim et al., 2016) or model-based inferences (Klein Goldewijk et al., 2011; Sohl et al., 2016).

The remainder of this data descriptor discusses the production, potential utility and uncertainty present in these new additions to HISDAC-US. Section 2 describes and showcases the data products. Section 3 discusses the underlying source data, the data processing, and introduces the validation datasets. Section 4 describes the types of uncertainty inherent in the data, and presents a thorough, systematic validation study against three different validation datasets. Section 5 describes data availability, and Sect. 6 provides some concluding remarks.

## 2  Main data products

Herein, we describe three novel time series of gridded, geospatial surfaces, representing long-term, spatially explicit building stock statistics for the conterminous U.S. over two centuries, at fine spatial and temporal detail. These datasets include two versions of the number of built-up property records (Uhl and Leyk, 2020a, b), derived from historical administrative and cadastral data sources that have been assembled in the ZTRAX database, aggregated into spatial bins (i.e., grid cells) of 250x250 m, at a temporal resolution of 5 years from 1810 to 2015, and a corresponding series of binary surfaces, indicating built-up areas (Uhl and Leyk, 2020c). The underlying binning grid is referenced to the USA Contiguous Albers Equal Area Conic projection (USGS version, SR-ORG:7480[2]). We derived the grid cell level aggregates from approximately 150,000,000 discrete point locations given in the ZTRAX database with each record representing a built-up property, of any usage type, including residential, commercial, industrial, recreational, or mixed building uses, among others. Importantly, a built-up property record may represent an ***individually-owned physical structure***, such as a single-family housing unit, an individually-owned factory

---

[1]Many of the global datasets mentioned here, use country-specific inputs in their training or modelling procedure.
[2]https://spatialreference.org/ref/sr-org/usa_contiguous_albers_equal_area_conic_usgs_version-2/

or commercially used building, a multi-unit building often in the form of large residential income housing units, or an office buildings owned by a single entity. A record may also represent an individually owned unit within a ***multi-owner structure*** such as a condominium unit or office unit within a larger physical structure. Records associated with multi-owner structures typically share the same geospatial location in the ZTRAX database. Thus, there are three meaningful ways to aggregate the ZTRAX built-up property records into grid cells:

1. Counting individual property records per grid cell, as a proxy variable for ***building units***. This count is reported in the first time series of datasets, the ***built-up property record (BUPR)*** surfaces.

2. Counting the unique locations of property records per grid cell, as a proxy variable for ***individual, physical built-up structures***. This count is reported in separate datasets, the ***built-up property location (BUPL)*** surfaces.

3. Indicating the presence / absence of at least one built-up property record per grid cell, as a proxy for ***developed land***, or built-up area. These binary surfaces are provided as separate datasets, the ***built-up area (BUA)*** surfaces.

We generated both BUPR and BUPL surfaces for each half-decade from 1810 to 2015, with each grid cell holding the count of records with a built-year attribute up to the year $T$. Moreover, we generated "contemporary" BUPR and BUPL datasets, summarizing the built-up property records and locations, respectively, regardless of their built-year attribute. Since we obtained the underlying ZTRAX data in early 2017, these contemporary layers reflect the BUPR and BUPL counts circa 2016. Likewise, we generated BUA surfaces for each half-decade, indicating the presence of at least one built-up property record per grid cell and year, as well as a "contemporary" BUA surface, reflecting developed land in 2016.

## 2.1 Built-up property record (BUPR) surfaces

The BUPR dataset series (Uhl and Leyk, 2020a) contains a gridded surface for each half-decade from 1810 to 2015, with each grid cell holding the count of records with a built-year attribute up to the respective year $T$. We highlight these gridded surfaces for selected years and regions in Fig. 1. Figure 1a shows the nation-wide BUPR surface for the conterminous U.S. in 2016. To illustrate both the spatial granularity and temporal coverage of the data, we visualized the directional sums of built-up property records for selected years along East-West and North-South cross-sections. The trends illustrate the well-known settlement patterns reflecting early colonial settlements in the North-East and subsequent expansion into the West and the South of the U.S.

The BUPR surfaces provide novel insights into regional, peri-urban and rural development, as shown in Fig. 1b for the Syracuse-Rochester region (New York). The map sequence documents both the existence and persistence of early, rural settlements, their growth in density over time, the simultaneous sprawl of towns and cities during the 20th century, and the emergence of settlements along the shorelines of the lakes in the center of the maps in the second half of the 20th century. At a more local scale, the BUPR time series enables the assessment of detailed long-term built-up development, as shown for the Eastern New Hampshire region in Fig. 1c, where settlement quickly expands and intensifies around the coastal town of Portsmouth, which

already exhibits a considerably large built-up area in 1810. Moreover, the potential of the BUPR surfaces for multi-temporal assessment of intra-urban building density variations can be seen in the video supplement.

## 2.2 Built-up property location (BUPL) surfaces

In residential neighbourhoods dominated by individually owned, single-family residential housing, the BUPL surfaces (building counts) (Uhl and Leyk, 2020b) closely resemble the BUPR surfaces (building unit counts). Differences are subtle and occur mainly in urban centres and regions where high-rise buildings and multi-unit buildings dominate the built environment. This difference is illustrated in Fig. 2, showing BUPR in 2015 for Denver, Colorado (Fig. 2a), and the corresponding BUPL surface (Fig. 2b). Differences become visible in a cell-by-cell ratio surface (Fig. 2c) where the Denver downtown area, dominated by high-rise buildings exhibits higher values.

## 2.3 Built-up area (BUA) surfaces

Built-up area (BUA) gridded surfaces (Uhl and Leyk, 2020c) represent a binary discrimination between built-up (value 1) and not built-up (value 0) areas, within 250x250 m grid cells, for each half-decade. The BUA surfaces are shown in Fig. 3(a-c) for selected U.S. cities, as well as the corresponding BUPR surfaces (Fig. 3(d-f) from which the BUA datasets have been derived through pixel-wise thresholding (i.e., a grid cell is considered built-up if BUPR>0). We also show grid cells where no built year information is available (Fig. 3c), which are provided as a separate dataset (Sect. 4.4.3). While the BUA surfaces have been employed for assessing long-term trends in land development (Leyk et al., 2020), and for the multi-temporal analysis of urban form (Uhl et al., 2020), they have not been published previously. These applications are evident from Fig. 3 which depicts the growth of cities, the increasing connectedness between urban cores and surrounding places (BUA, Fig. 3a-c), intra-urban density variations across space and time (BUPR, Fig. 3d-f) and the ability for these surfaces to characterize urban settlement trends. For example, the BUA surface for 1915 (Fig. 3a) highlight, with unprecedented spatial detail, the well-known disparity between early-developing North-eastern cities and the slower urban development of the South (see video supplement for a corresponding animation). Thus, these visualizations highlight the empirical value of these surfaces in assessing heterogeneity in urban growth over long temporal extents and with (currently) unparalleled spatial detail (see video supplement). While advanced GIS practitioners would be able to derive the BUA surfaces from the BUPR / BUPL datasets, we provide them as a separate dataset, to facilitate the use for applications where binary built-up / not built-up differentiation is sufficient. Moreover, the BUA surfaces are assumed to be least affected by uncertainties in the ZTRAX data (see Sect. 4.1).

## 3 Data and data processing

## 3.1 Source data and data processing

The ZTRAX database is based on existing cadastral data sources and contains more than 400 million data records (Zillow Inc., 2016), out of which around 150,000,000 contain spatial information, while the remaining 250 million records represent

transactional records (e.g., detailed information on property sales, etc.), and other aspatial data tables, as well as the database history. This database is available to the authors via a data share agreement and is used as a basis to derive publicly available datasets, enabling scientists to benefit from the spatial, temporal, and semantic richness of ZTRAX. The raw ZTRAX database consists of around 2,500 state-level text files of a total volume of 1.4 TB, with each file representing a table of the original database. The data tables are thematically split into three major groups (i.e., contemporary and historical assessment data, and transaction data) (Fig. 4a). We used the Feature Manipulation Engine (FME; Safe Software Inc., 2020) to import these files into a set of SQLite relational databases (SQLite, 2020). Using SQL queries and the ESRI ArcPy (ESRI, 2019) python package we retrieved relevant attributes and extracted them as geospatial vector datasets into ESRI Geodatabases. Geometries were generated using the geospatial information contained in ZTRAX (i.e., geographic coordinates), representing address points or cadastral parcel centroids given as geographic coordinates in NAD1927 datum. These geolocations have been generated by Zillow Inc. using geocoding and spatial refinement techniques. We then imported each of the 3000+ county-level geospatial vector datasets into GeoPandas (Jordahl et al., 2020) data frames and projected all records that indicate the presence of a built-up structure into Albers Equal Area Conic projection for the conterminous United States (CONUS) (SR-ORG:7480). More specifically, we excluded records of land use type "vacant land". Based on the built year attribute, we generated temporal slices of the data points (in 5 a increments, i.e., all records built-up between $T$ and $T-5a$ and computed 2d-histograms using NumPy python package (Oliphant, 2006), with histogram bins derived from the underlying 250x250 m grid covering the CONUS. This approach allows for an efficient spatial binning of the vast amounts of data points. Using temporal slices of 5 years kept the total number of data points to a minimum and significantly reduced the overall processing time. For the BUPL surfaces, which contain unique locations of property records within each grid cell, duplicate coordinate pairs were removed prior to the spatial binning step. The resulting 2d-histogram arrays were then exported as GeoTIFF using the Geospatial Abstraction Data Library (GDAL; GDAL/OGR contributors, 2020). Lastly, in order to obtain the total counts of built-up property records and locations for each half-decade $T$, all temporal slices from the year 1810 to $T$ were added up cell-by-cell. The complete processing of all 150 million data records took around 2.7 days and is illustrated in Fig. 4b.

## 3.2 Validation data

We conducted an extensive validation study of the generated BUPR and BUPL surfaces against three different validation datasets, and across different domains. The validation datasets include contemporary building footprint data for the CONUS (Microsoft, 2018), an integrated, multi-temporal database of building footprint data and cadastral parcel records (Uhl et al., 2016), as well as historical U.S. census housing counts (Manson et al., 2019) (see Table 1 for details). Moreover, a U.S. county boundary dataset (US Census Bureau, 2017), a U.S. census-designated places boundary dataset (US Census Bureau, 2017), and USDA rural-urban continuum codes (USDA Economic Research Service, 2020) were used for stratified validation. These datasets are described in detail in the following subsections.

### 3.2.1 Contemporary US-wide building footprint data

We used Microsoft's U.S. building footprint data (MSBF), which have been generated from Bing maps imagery (i.e., a compilation of different airborne and spaceborne remote sensing data sources; Microsoft, 2018) using a deep learning based computer vision algorithm. This database contains more than 125,000,000 building footprints and is available in GeoJSON format. According to the data producers, this dataset is highly accurate (i.e., precision of 0.993, recall of 0.935; Microsoft, 2018) and thus represents the most reliable, recent, and complete data source of building footprint data in the U.S. We used FME software to convert the GeoJSON data into ESRI File Geodatabase format and aggregated these data into grid cells in analogy to the data processing step as described in Sect. 3.1. This approach allowed us to create a U.S.-wide, highly reliable reference building density surface, referred to the grid cell area of $0.0625$ $\text{km}^2$, approximately temporally referenced to the year 2016, and compatible to the BUPR and BUPL surfaces (i.e., using the same underlying grid). This surface and the underlying building footprint data are shown in Fig. 5a.

### 3.2.2 Multi-temporal building footprint data

While MSBF data covers the whole CONUS, it is available for one point in time only. To evaluate the agreement of BUPR and BUPL surfaces with reference measures of building density over time, we used an integrated data product of building footprint data and cadastral parcel records. Built-year information from the cadastral parcel data (Fig. 5b) was transferred to the (typically LiDAR-derived) building(s) contained within the parcel (Fig. 5c; Uhl et al., 2016). This database was used previously for validation studies of the HISDAC-US BUI surfaces (Leyk and Uhl, 2018a) and remote sensing derived settlement data (Leyk et al., 2018; Uhl et al., 2018), and tested as training data for remote sensing based urban change detection (Uhl and Leyk, 2020e). By querying the building footprints by their built-year attribute, this database enables the creation of granular snapshots of built-up areas for user-specified points in time. The geographic coverage of this database is constrained to 30 U.S. counties, where there is publically available parcel data on built year (see Table B1). Based on this multi-temporal building footprint database (herein referred to as MTBF30), we created time slices of building footprints and generated corresponding gridded building density surfaces for each half-decade, as shown in Fig. 5d,e.

### 3.2.3 Multi-temporal U.S. Census housing statistics

As a third validation dataset, we employed historical U.S. census housing unit counts. While for recent census years (e.g., 1990-2010), housing unit counts are available at very fine spatial granularity (i.e., census tract and finer), in earlier years such data is available at the county level only. We used historical county boundaries and housing unit counts obtained from the National Historical Geographical Information System (NHGIS; Manson et al., 2019) for all available years, i.e., 1890–1940, and 1970–2010. These county-level counts are shown in Fig. 5f for selected years.

### 3.2.4 Rural-urban continuum classification data and U.S. census-designated place boundaries

Uncertainty in many geospatial datasets increases from urban towards rural settings (see e.g., Smith et al., 2002; Wickham et al., 2013; Leyk et al., 2018). In order to examine if the ZTRAX data and the derived HISDAC-US data products exhibit this trend, we examined uncertainty trajectories across the rural-urban continuum, as modelled by the U.S. Department of Agriculture (USDA) rural-urban continuum codes (RUCC 2013; Butler, 1990). These codes assign a degree of "rurality" to each U.S. county, on a scale from 1 (most urban) to 9 (most rural), based on proximity to cities of certain population sizes (see Fig. 5g). Due to the lack of RUC codes covering the entire study period (i.e., 1810 – 2016) we used the most recent RUCC definition from 2013 for stratified assessment of the 2016 data only (Sect. 4.2.2). Moreover, we assume data uncertainty to vary between incorporated places (i.e., villages, towns, cities) and more fragmented and dispersed rural settlements. To account for this uncertainty, we used 2010 U.S. census-designated place boundaries (US Census Bureau, Department of Commerce, 2016, herein referred to as "census places") to analyse uncertainty separately within county boundaries (i.e., including rural settlements that are not incorporated into a census place (see Fig. 5h) and within census place boundaries only (see Fig. 5i).

### 3.2.5 Data on public housing and buildings

As publicly-owned buildings are mostly not contained in the ZTRAX dataset, we employed several auxiliary datasets to quantify the effects of these omissions. These auxiliary datasets include (a) United States Geological Survey (USGS) national structures dataset (NSD, USGS National Geospatial Technical Operations Center, 2016), (b) U.S. Department of Housing and Urban Development (HUD) data on public housing (U.S. Department of Housing and Urban Development, 2019), and (c) public amenities from Open Street Map (OpenStreetMap contributors, 2020) (see Appendix C).

## 4 Data uncertainty and validation

The BUPR and BUPL datasets suffer from several types of uncertainty, mainly inherited from the underlying ZTRAX data. These types of uncertainty can broadly be categorized into three groups: ***Data incompleteness, locational uncertainty***, and ***quantity disagreement***. ***Data incompleteness*** encompasses incomplete geographic coverage (e.g., data gaps) of the ZTRAX data, as well as attribute incompleteness, resulting from missing attribute values in the underlying ZTRAX database, and the omission of public properties and buildings in ZTRAX (Appendix C). We analysed data incompleteness at the county, census place, and grid cell level (Sect. 4.1). Moreover, the ZTRAX data suffer from a certain ***survivorship bias***, resulting from lacking information on building teardowns, and potentially inconsistent records on building replacements (Sect 4.1). ***Locational uncertainty*** results from uncertainty in the geospatial information reported in ZTRAX, and includes issues due to spatial generalization (Sect. 4.2.1) and low positional precision (Sect. 4.2.2 and 4.2.3). Lastly, we used our validation dataset to assess ***quantity disagreement*** in the BUPR and BUPL densities, including (systematic) under- and over-estimation (Sect. 4.3). At this point, it is worth noting that the systematic underestimation of BUPR and BUPL towards early years may be a result of lacking information on building teardowns and replacements in ZTRAX (see Sect. 4.3).

Herein, we expand on previous analyses of these uncertainties (Leyk and Uhl, 2018a) to provide a more in-depth assessment of these shortcomings and their implications for data users. More specifically, we employ additional validation datasets and explicitly assess these uncertainty types across time, and across the rural-urban continuum.

## 4.1 Data incompleteness

Data incompleteness consists of two components: (a) incomplete geographic coverage of the ZTRAX data (i.e., data gaps), and (b) incomplete coverage of specific attributes in the ZTRAX database. The **geographic coverage** of the ZTRAX data extends across 3,026 out of 3,108 counties in the CONUS. The remaining 82 counties do not have any geospatial ZTRAX data records (Fig. 6a,b). These counties correspond to 2.5% of the CONUS area and were inhabited by 0.82% of the US population in 2010. 73% of these counties are classified as "non-metropolitan" (i.e., RUC codes 4 to 9), according to the USDA rural-urban classification in 2013 (USDA Economic Research Service, 2019, 2020). An additional source of incomplete coverage arises from **thematic limitations** in the ZTRAX data, i.e., the omission of publicly-owned buildings. Many big cities have public housing projects, which may be omitted from the ZTRAX records. We quantified the effects of the omission of publicly-owned buildings using three auxiliary data sources (see Appendix C).

Moreover, we analysed the **built-year attribute coverage**, which is the most relevant attribute for the creation of the multi-temporal BUPR and BUPL surfaces. The built-year attribute exhibits high levels of completeness, with notable exceptions including states in the Northern Midwest, Vermont, Louisiana, and New Mexico (Fig. 6a). A county boundary shapefile containing the county-level summary statistics underlying Fig. 6a was published and is available to data users (Leyk and Uhl, 2018d). We computed the same completeness statistics within census place boundaries (Fig. 6b) and observe higher levels of built-year attribute completeness in Western and Midwestern states. This result indicates that built-year attribute missingness is likely to affect records in unincorporated, spread-out rural settlements, rather than those in urban areas or census-designated places such as towns or villages. We provide a gridded dataset flagging grid cells without any built year information (Fig. 6c, see also Fig. 3c, Sect. 4.4.3) that allows for excluding the respective areas, constituting approximately 2.7% of the CONUS landmass. The previously made observation is confirmed in the boxplots shown in Fig. 6d, indicating, on average, higher levels of built-year completeness within census place boundaries than within county boundaries. In addition to that, Fig. 6d reveals clear trends of increasing built-year incompleteness from urban to rural counties.

Importantly, the ZTRAX data and derived datasets suffer from a **survivorship bias**, or selection bias, that increases towards early points in time, and manifests in omission errors both affecting locational uncertainty over time (Sect. 4.2.3) and quantity agreement over time (Sect. 4.3.1 and 4.3.3). This bias is introduced by lacking consistent information on building demolitions and replacements in the ZTRAX data, as well as the absence of information about properties existing prior to building replacements. This reasons for this bias can be three-fold: 1) Demolished buildings that existed in the past, without being replaced by a contemporarily existing structure are not contained in the data. 2) The built year information contained in ZTRAX at a given location typically represents the year when the first structure at that location was built, but may also indicate the year of a replacement, as empirical tests have shown. Thus, the part of a structure's life span prior to the replacement may not be measured by our data. 3) Finally, the number of property records associated with a given location and built year may have been

different in the year when the first structure was built. While the former two components of this bias would result in omission errors, the latter component could result in both a commission error (e.g., if the built year associated with a multi-owner structure in fact represents the built year of a single family home that has been replaced) or in an omission error, if small, individual properties have been replaced by large, single-owner structures. While these individual components of survivorship bias are difficult to assess in detail, the assessments in Sect. 4.2.3, 4.3.1, and 4.3.3 allow us at least to quantify the upper bounds of the effects introduced by this bias. Here, it is worth noting that the binary BUA surfaces are expected to be least affected by the survivorship bias, as they are based on the presence of ZTRAX records, independently from the quantity of records per grid cell.

## 4.2 Locational uncertainty

We group locational uncertainties in the ZTRAX data that propagate into the derived HISDAC-US surfaces into two main categories: (a) Locational uncertainty due to spatial generalization of the geospatial information in ZTRAX and (b) positional imprecision of the spatial information (i.e., geospatial coordinates deviating from actual building locations). The latter component may be affected by the geocoding quality and by the spatial refinement methods used by Zillow Inc. We developed several visual and analytical methods to assess and quantify these uncertainties, and we provide additional uncertainty surfaces accompanying the BUPR and BUPL datasets (Sect. 4.4).

At this point, it is important to describe some issues related to the geospatial locations reported in ZTRAX. In urban, single-family residential neighbourhoods, geospatial locations are typically derived from cadastral parcel centroids. Parcel sizes are typically similar in size to the buildings within parcels, and thus, the locations given in ZTRAX are likely to spatially coincide with the location of the building (Fig. 7a). In peri-urban and rural, agricultural settings, where parcels are often large, the parcel centroid may be far from the actual building location and thus, may provide a less precise estimate of the actual location of the built-up structure in question (Fig. 7b). This precision also applies to cases where address points are used. Address points typically represent the location of a building snapped to the road median, as an approximate location of the mailbox, in cases where buildings are located far off the road (see Fig. 7c). These issues may potentially result in locational precisions below the parcel level (see also Nolte, 2020). While these effects are expected to be partially mitigated by the 250x250 m grid cell aggregation, our BUPR, BUPL, and BUA surfaces may not accurately reflect the location of actual built-up structures, particularly in rural areas.

Moreover, due to the ZTRAX data model, built-up property records reflect legal ownership. These records may represent an individually-owned built-up structure, such as single family residential buildings, or an individually owned multi-family building (residential income, i.e., apartments). If housing units within physical structures are individually owned, each unit is represented as an individual property record in the ZTRAX database (multi-owner records, i.e., condominiums). This designation also applies to residential communities, which may encompass multiple physical structures (multi-address records). This peculiarity of the ZTRAX data model may lead to multiple overlapping records at the same location. We refer to these cases as "multi-record locations" (represented in the BUPL surfaces), and their associated records as "multi-records". If such multi-records are encountered in regions characterized by high-rise buildings (cf. Fig. 2c), their locational uncertainty is low,

since the properties (i.e., building units) represented by these records are, in fact, stacked on top of each other. However, there are cases when such multi-records are used for spatially more spread-out structures or complexes, such as mobile home parks (Fig. 7d) or planned communities (Fig. 7e). As illustrated in these examples, the reported locations of these multi-records may deviate considerably from the actual location, and thus, introduce positional error in the gridded BUPR and BUPL surfaces. Moreover, densities at those locations can be exorbitantly high. While ZTRAX contains a considerable number of such locations (see Sect. 4.2.1), there are, to a much lesser extent, multi-record locations as a result of "pseudo-locations". These pseudo-locations were likely assigned as rough location estimates for built-up property records in places where detailed spatial information was not available during the original database creation. Such an example is shown in Fig. 7f, where the highlighted multi-records likely represent nearby properties.

The illustrations shown in Fig. 7, aim to raise awareness among data users that positional accuracy can be low in areas with mobile home parks, sprawling residential housing, apartment buildings or condominiums, typically represented by multi-record locations. Fig. 7 also illustrates that pseudo-locations may be the reason for extreme BUPR counts in sparsely, rural regions or in developing areas. While the effects of spatial generalization cannot be quantified without manual checks against aerial imagery, or the use of rarely available volumetric building data, we conducted a spatial analysis of these multi-record locations (see Sect. 4.2.1). This analysis provides additional insight into how and to what degree multi-record locations and the associated potential positional errors may bias the generated BUPR and BUPL surfaces (see also Appendix D).

### 4.2.1   Analysing spatial generalization effects

Out of 117.5 million built-up property records in CONUS in 2016, there are 89.5% referenced to unique spatial locations, and 10.5% share the geospatial location with at least one other record (i.e., multi-records). Among the 101.7 million built-up property locations, 96.7% contain a single record, and only 3.3% contain two or more records (i.e., multi-record locations). From these 3.3%, a proportion of 6.7% of the multi-record locations contain built-up records that include mobile home parks and other residential income properties, and 27.9% of the multi-record locations contain usage types related to office space, planned communities, or residential condominiums. Thus, the potential, positional inaccuracies discussed above affect only a small proportion of the data, as these numbers indicate.

For example, Fig. 8a shows the BUPR 2016 surface for Denver, Colorado, and Fig. 8b shows only the grid cells that contain at least one multi-record location. It is not surprising that these grid cells are mainly found in the downtown area (map center), which is dominated by high-rise commercial buildings and office condominiums. Additionally, we used a land use type attribute reported for each record in the ZTRAX database (McShane et al.) to analyse the usage type at multi-record locations. To do so, we flagged multi-record locations involving office or residential condominiums, and large residential income usage (e.g., mobile home parks, large apartment complexes). Grid cells with multi-record locations not involving office, residential condominiums or mobile home parks are shown in Fig. 8c, and d, respectively. Most of these multi-record locations hold two or very few multi-records, and likely represent parcels with multiple buildings, e.g. commercially or industrially used parcels. A few, spatially isolated grid cells in peri-urban areas (Fig. 8d) indicate multi-record locations holding higher numbers of multi-records and may represent pseudo-locations in developing areas, which will likely be refined in future ZTRAX database

versions. However, multi-record locations containing extremely high numbers of records are very rare and follow a rank-size distribution, as the rank-size plots in Fig. 8e suggest. Users are able to mitigate the effect of these locations using the accompanying positional uncertainty surface (Sect. 4.4.1). See Appendix D for further analyses of multi-record locations.

### 4.2.2 Positional accuracy across multiple spatial resolutions

Due to the nature of locational information in ZTRAX, the created BUPR and BUPL surfaces do not necessarily reflect the precise locations of physical built-up structures, as previously discussed. Lower levels of precision due to large parcel sizes (Fig. 7b,c) and spatial generalization effects introduced by certain types of multi-record locations (Fig. 7d-f, Sect. 4.2.1) generate positional uncertainty in the resulting surfaces. To quantify positional accuracy of the 2016 BUPR, BUPL, and BUA surfaces, we conducted cell-by-cell map comparison against the reference surface generated from the MSBF data (Sect. 3.2.1).

While positional agreement assessment using map comparison techniques is a commonly applied method in remote sensing and related sciences, it assumes semantic compatibility between reference data and data under test, i.e., the geographic process measured by both datasets should be identical. In our case, we compare building outlines to locations derived from parcel centroids or address points, possibly resulting in spatial disagreement between the (gridded) test and reference data, even though both datasets are in agreement (i.e., ZTRAX location and building footprint are within the same parcel boundaries).

Hence, spatial disagreement (i.e., false positive or false negative instances) is assumed if we can rule out that the disagreement is induced by spatial offsets due to different semantics (i.e., parcel centroid / address point vs. building footprint) and spatial granularity (i.e., discrete point vs. polygon) between underlying test and reference data. Our method models the probability that positional disagreement is induced by such spatial offsets, and is based on the "contemporary" $BUA_{2016}$ surface (Fig. 9a) which is compared against a binary built-up presence surface derived from MSBF data (Fig. 9b). This multi-scale approach

(see Appendix E for details) quantifies agreement at multiple spatial aggregation levels (i.e., cell sizes), and generates a surface of offset-induced misclassification probability (Fig. 9c-f).

We established confusion matrices for each aggregation level, within county and census place boundaries, and assessed the agreement separately for each county-level rural-urban continuum code (RUCC) within county and place polygons (see Fig. 5h,i), excluding counties without ZTRAX data coverage (Fig. 6a,b). This approach allows for extracting positional agree-

ment measures (i.e., precision / User's accuracy, recall / Producer's accuracy, and F-measure) across aggregation levels and across the rural-urban continuum, both within census place boundaries and overall (within county boundaries, i.e., including scattered rural settlements outside of census places). There are high levels of precision across all RUCCs, particularly within census place boundaries (i.e., >0.89; Fig. 10a). Recall shows slightly lower values in rural regions (i.e., RUCCs 6-9), but also in urban regions (i.e., 0.88) which is likely due to the omission of publicly-owned buildings in ZTRAX. When evaluating

agreement using county boundaries (i.e., including settlements not incorporated into census places, such as dispersed, rural settlements, Fig. 10b), we observe a drop in accuracy, in particular for recall in rural areas. This decline indicates lower levels of completeness of ZTRAX in predominantly rural places but may also be related to inaccuracies in the MSBF data (see Appendix F). All accuracy measures increase with increasing spatial aggregation level, in particular in rural areas for aggregation factors 2 and 4 (corresponding to 500m and 1000m, respectively), where offsets between underlying ZTRAX locations and

building footprints may be large (cf. Fig. 9e). In these cases, the spatial aggregation method is particularly effective and likely provides a more unbiased accuracy estimate.

Moreover, we examined how the offset-induced misclassification probability changes across the rural-urban continuum. As illustrated in Fig. 10c, which is based on calculations within county boundaries, we observe that the proportion of false positives and false negatives with high offset-induced misclassification probability increases steadily, from 24% of the true positives in
urban counties (RUCC 1), to 53% in intermediate counties (RUCC 5), to 82% in the most rural counties (RUCC 9). Based on these observations and given the spatial offsets between ZTRAX data and validation building footprint data, we assume that offset-induced bias is the main cause for low recall measures in rural settings. Hence, the accuracy trajectories for aggregation levels 2 or even 4 (Fig. 10a,b) are likely to show a more realistic picture of the agreement between the BUPR, BUPL surfaces and the validation data.

### 4.2.3   Positional accuracy over time

While the previous assessment illustrates accuracy trajectories across the rural-urban continuum, it is based on the contemporary built-up areas (i.e., derived from the BUPR 2016 surface) and does not assess accuracy variations over time. Since our previous work on temporal accuracy trajectories (Leyk and Uhl, 2018a) has not differentiated between predominantly urban and rural places, we fill this gap by computing positional agreement measures of the binarised, multi-temporal BUPR surfaces
against the reference surfaces generated from our MTBF30 database, for each half-decade, and separately for high-density and low-density counties (see Sect. 3.2.2). Since this reference database covers 30 counties in the U.S., and thus, represents a rather small sample, we computed county-level building densities based on the reference data. Using the 75th percentile of building density measures for each point in time as a threshold, we separated the 30 counties into counties of predominantly low and high built-up density (Table B1), rather than using the USDA RUC codes temporally referenced to 2013. Results are
shown in Fig. 10d,e for predominantly rural and urban counties, respectively, indicating high levels of precision since the early 1800s, whereas recall drops almost logarithmically when going back in time. This indicates higher levels of omission errors for structures established prior to 1900. However, it is also affected by larger positional offsets between ZTRAX and building data for older structures. Previous work included a multi-temporal accuracy assessment across different levels of spatial aggregation (Leyk and Uhl, 2018a) and showed that, for a spatial aggregation level of 1,250 m, recall values in 1850 increase to over 0.75.
Moreover, accuracy levels are slightly lower in predominantly rural counties (Fig. 10d) than in urban counties (Fig. 10e).

### 4.3   Assessing quantity agreement

Lastly, we assessed the quantity agreement of the counts reported in the BUPR and BUPL surfaces to our validation datasets at different spatial granularity and across different domains: (a) Agreement over time between county-level housing unit counts obtained from the U.S. census (Sect. 4.3.1), (b) agreement across the rural-urban continuum at grid-cell level building counts
generated from the MSBF dataset (Sect. 4.3.2), and (c) agreement over time against our MTBF30 database (Sect. 4.4.3). Since the validation datasets are based on different measurements, but are, to a certain degree, semantically coherent to the BUPR

and BUPL surfaces, we expected certain levels of disagreement when comparing these counts but high levels of association and correlation over time.

### 4.3.1 Multi-temporal quantity agreement against census-based housing statistics

We visualized the distributions of census-based county-level housing unit counts and built-up property counts, aggregated to county boundaries of the respective census years (Fig. 11), for 1890 – 1940 and 1970 – 2010, separately for counties of predominantly rural (Fig. 11a) and urban character (Fig. 11b). We did this rural-urban stratification based density percentiles for each point in time, as described in Sect. 4.2.3. We observe very similar trends of built-up properties and housing units over time, with census housing units systematically exceeding the ZTRAX-derived built-up property counts. This difference

may stem from residential income housing, such as large rental-based apartment complexes, that appear as a single property record in ZTRAX, but are represented as multiple housing units in the census data. While this explains the differences in urban counties (Fig. 11b), the deviations in rural counties (Fig. 11a) may be a result of higher omission errors (i.e., lower recall values) in the ZTRAX data in earlier points in time (cf. Fig. 10). Agreement trends derived for BUPL surfaces look largely similar as indicated by the time series of Pearson's correlation coefficients ( Fig. 11e). The correlations are high for both BUPR and

BUPL in high-density counties (i.e., >0.8 since the year 1900) but exhibit lower levels of agreement in low-density counties, due to higher omission errors in the ZTRAX database in rural settings, where data tend to be less reliable and cadastral data acquisition may not be a priority. Moreover, we observe an increasingly linear relationship over time between BUPR / BUPL and census-based housing unit counts (Appendix Fig. G1 b,c,d).

### 4.3.2 Quantity agreement across the rural-urban continuum

The relationships at the grid-cell level between the BUPR 2016 surface and the reference surface derived from MSBF data (Sect. 3.2.1) show a clear trend across the rural-urban continuum (Fig. 11c). While most grid cell pairs are found near the main diagonal in these scatterplots in urban counties (RUCC 1), a second (lower) branch is visible. This branch results from grid cells of high BUPR, but low reference building counts, likely representing high-rise buildings, large apartment buildings and office condominiums. Moreover, this progression illustrates the density decline from urban towards rural settings. The corresponding

robust regression results (Huber et al., 1973, see also Appendix Fig. G1) indicate linear relationships with slope values around 1.0 for both BUPR and BUPL surfaces. The slope for the BUPR distribution is lower (0.68) in rural counties (RUCC 9), likely a result from few, but very high-valued multi-record locations, potentially representing "pseudo-locations" occurring in rural regions (cf. Sect. 4.2.1). In comparison to the BUPR regression lines, the slope coefficients from the BUPL-based regression models are consistently closer to 1.0, indicating slightly stronger associations between built-up property locations

and building counts. $R^2$ values of these regressions are consistently very high across all RUCCs (Appendix Fig. G1h), as well as the correlation coefficients for each half-decade. They exhibit slightly higher correlations for BUPL than BUPR, with slight drops in highly urban and highly rural strata (Fig. F1e).

### 4.3.3 Quantity agreement over time at the grid cell level

BUPR / BUPL and gridded building footprint counts derived from the MTBF30 database (Fig. 11d) show a general increase
in both, building and built-up property record counts at the grid cell level across the 20th century. Counts increases notably
during the first half of the 1900s (i.e., densification), while growth in built-up area after 1950 occurred increasingly also in
form of suburban expansion (Leyk et al., 2020). These relationships are highly linear across all points in time. Similar to the
observation made in RUCC 1 counties (Sect. 4.3.2), the surfaces in the year 2000 show an emerging accumulation of grid cells
with high BUPR values, but low building counts (likely high-rise buildings, etc.). In addition, larger numbers of data points
above the main diagonal appear since 1950, i.e., where reference building counts exceed the number of property records. This
result may be attributed to some underestimation in the ZTRAX database but is more likely to be a result of increasing numbers
of properties with several, physically separated buildings, such as garages, sheds, or carports contained in the reference building
database. These data points also cause the BUPL regression line slopes of >1.0, which we do not observe in the MSBF-based
scatterplots (Fig. 11c). This observation is likely an effect of the low sample size in the multi-temporal building database (1%
of U.S. counties) as compared to the MSBF data coverage, and the under-representation of high-rise buildings located in highly
urban settings.

Corresponding correlation time series (Fig. 11f) reveal several interesting insights. First, correlation levels over time are
fairly high back to 1850 and drop below 0.8 only prior to that. Second, correlation between building counts and BUPL are
consistently higher over time than for BUPR, indicating that changes in the number of buildings over time are reflected better
in the BUPL surfaces than in BUPR, likely a result of multi-record locations holding large numbers of property records. Third,
correlations are higher in the low-density counties than in high-density counties, and are lowest for BUPR in high-density
counties. This trend is likely due to higher numbers of multi-apartment buildings in high-density areas as compared to low-
density areas, resulting in larger deviations of BUPR from the number of physical built-up structures within grid cells. The
higher correlations in low-density counties are surprising, since we found low correlations to census-based housing unit counts
in rural (low-density) counties (Fig. 11e). Moreover, stable slope values and high $R^2$ values over time since 1850 imply a
strongly linear relationship between BUPR / BUPL and MTBF30 data (Appendix Fig. G1j). These observations reveal that
the BUPR, and BUPL surfaces hold great potential to describe changes in the built environment across different settings, but
show different associations with housing trends as reported and defined by the census over time, particularly in rural settings.
A quantitative assessment of the differences between BUPR / BUPL counts and the reference data counts can be found in
Appendix Fig. G2.

### 4.4 Accompanying uncertainty surfaces

To allow users to mitigate and reduce the effects of locational uncertainty inherent in the BUPR, BUPL and BUA surfaces,
we provide three accompanying uncertainty surfaces at a spatial resolution of 250 m (Uhl and Leyk, 2020d). These surfaces
are: (a) a "multi-record count surface", as a measure of potential positional uncertainty due to spatial generalization of the
underlying ZTRAX data records (Sect. 4.2.1), (b) a positional reliability surface, containing the agreement / disagreement type

for each grid cell, obtained by map comparison against the MSBF-derived reference surface (Sect. 4.2.2), and (c) a built year missingness surface, flagging grid cells containing built-up properties but no built year information (Fig. 5c).

### 4.4.1 Multi-record count surface

The multi-record maxima surface contains for each grid cell in the CONUS, the maximum number of built-up property records with the same geolocation. This count does not include any residential income or office / residential condominium land use type, as shown in Fig. 8d. Extreme grid cell values in this gridded surface may indicate the presence of "pseudo-locations" (see Sect. 4.2.1). The data user can decide how to employ this surface to mask out locations in question by applying a suitable threshold.

### 4.4.2 Positional reliability surface

The positional reliability surface is a simplified version of the probabilistic agreement/disagreement surface shown in Fig. 9c-f, containing three classes (i.e., true positive, false positives, false negatives) resulting from map comparison against the MSBF data. This surface enables the data user to identify grid cells that represent commission / omission errors with respect to MSBF data, such as sub-county level data gaps not captured in the county-level uncertainty statistics available for the HISDAC-US (Leyk and Uhl, 2018d). Such sub-county level data gaps are, in part, due to the previously described omission of publicly-owned buildings in ZTRAX (cf. Appendix C). Here, it is worth noting that many cities provide geospatial datasets indicating the location of their public-housing buildings (see e.g. NYC Housing Authority, 2020; City of Los Angeles Controller's Office, 2017) at least for contemporary periods, and such data could be used to quantify and mitigate these specific omission errors in detail. Moreover, positional uncertainty (i.e., deviations from actual building locations) may be introduced by imprecise geolocations as a result of Zillow's geocoding and spatial refinement strategy. Besides this positional reliability surface derived from the MSBF data, we refer the reader to previously published positional uncertainty surfaces that take into account the parcel size of a ZTRAX record, and the distance of a given geolocation to the grid cell edges (Leyk and Uhl, 2018b).

### 4.4.3 Built year missingness surface

The built year missingness surface flags grid cells that contain built-up property records, but no built year information, allowing data users for excluding regions where changes over time cannot be directly measured. This binary "no-built-year" (NBY) surface is, in similar form, contained in the FBUY surface (grid cells of value 1; Leyk and Uhl, 2018a, c). While this binary surface allows for excluding grid cells without any temporal information, users may be interested in excluding grid cells based on certain proportions of locations (i.e., BUPR) without built year information. To do so, we refer to our previously published dataset "TPixMiss" (Temporal pixel missingness) containing the number of BUPL without built year per grid cell (Leyk and Uhl, 2018b).

# 5 Conclusions

This data descriptor introduces a novel fine-grained building dataset that spans two centuries of development history in the United States. By providing unique insight on the long-term dynamics of urbanization and the built environment, the spatiotemporal richness of this dataset vastly expands the opportunities to study land-use and land-cover change over extended periods of time. Not only do these geospatial gridded surfaces enable the measurement of physical building density through time, but can also be flexibly integrated with standard demographic data sources like the decennial census. While no reference data can fully validate a data source of this scale and scope, we conducted cross-comparisons of the counts provided in the BUPR and BUPL surfaces to a variety of validation datasets. While our exercises reveal generally high levels of reliability, there is substantially higher uncertainty in our observations from before 1850. The absence of information on building teardowns or replacements in the ZTRAX data is one plausible explanation for this inconsistency. In future work, we will test strategies to quantify these uncertainties in detail by employing auxiliary data sources. This will potentially enable us to provide refined uncertainty estimates of even corrected datasets. Preliminary tests have shown promising results and that this issue has only minor effects on analytical outcomes (Uhl et al., 2020). This said, by utilizing our uncertainty estimates, data users can incorporate uncertainty into their analyses and mitigate data discrepancies. These new data products provide an unprecedented baseline for the modelling of spatio-temporal phenomena related to urbanization, land-use transitions and even demographic change (see Leyk et al., 2020). Moreover, many of the challenges highlighted in this article can be tackled through the development of cutting-edge data imputation strategies. Taken together, this dataset will enable predictive models to learn from the past, to better predict future environmental, social, or demographic scenarios. Lastly, these BUPR and BUPL gridded datasets are the newest contribution to our expanding HISDAC-US compilation, which is making unique industry-generated data derivatives available to scientists within and beyond the earth systems research community.

*Code availability.* Source code for data extraction, processing, and analysis is available from the authors upon reasonable request.

*Data availability.* The built-up property record (BUPR, Uhl and Leyk, 2020a), built-up property location (BUPL, Uhl and Leyk, 2020b), and built-up area (BUA, Uhl and Leyk, 2020c) gridded surface time series are available as part of the Historical Settlement Data Compilation for the U.S. (HISDAC-US) at https://dataverse.harvard.edu/dataverse/hisdacus. The data are provided as geospatial raster layers, at a spatial resolution of 250x250 m, one layer for each 5-year period, from 1810 to 2015. Gridded datasets are spatially referenced using the Albers Equal Area Conic projection for the CONUS (SR-ORG:7480), and data are available in GeoTIFF format using LZW data compression. The uncertainty surfaces accompanying the BUPR, BUPL, and BUA surfaces are the no-built-year (NBY) surface, the multi-record maxima surface, and the positional reliability surface and are also available as gridded datasets (Uhl and Leyk, 2020d), at identical spatial resolution and reference, in the HISDAC-US data repository. The first built-up year surface (Leyk and Uhl, 2018c), the built-up intensity surfaces (Leyk and Uhl, 2018b), and county-level uncertainty statistics (Leyk and Uhl, 2018d), as described in Leyk and Uhl (2018a) are also accessible at https://dataverse.harvard.edu/dataverse/hisdacus. See Table 2 for an overview of the different data products.

*Video supplement.* We provide a supplementary video file highlighting the BUA and BUPR surface time series. This video shows (a) the BUA surfaces (i.e., developed and non-developed grid cells), (b) BUPR surfaces (i.e., number of built-up property records per 250 m grid cell), and (c) changes in built-up areas for 35 selected U.S. cities, for each half-decade from 1810 to 2015. These changes represent newly built-up grid cells, and are obtained from cell-by-cell differences of the BUA surfaces for two subsequent points in time. Changes are shown for moving
time intervals of 30 a, to better highlight the medium and long-term development trends. The cities are arranged in an approximate geographic space (i.e., North-eastern cities are shown in the upper right part of the array). The video is available at https://doi.org/10.5446/48115.

*Author contributions.* S.L. made the data accessible and secured funding. J.U. processed the provided ZTRAX data into analysis-ready data formats. J.U. generated the BUPR and BUPL surfaces, as well as the BUI and FBUY datasets, and the associated uncertainty datasets in the HISDAC-US data compilation. S.L. and J.U. designed the validation study. J.U. visualized the data, carried out the validation experiments and visualized the results. J.U., S.L., C.M., A.B., D.C., and D.B. wrote the paper.

*Competing interests.* The authors declare no conflict of interest.

*Acknowledgements.* Funding for this work was provided through the Humans, Disasters, and the Built Environment program of the National Science Foundation, Award Number 1924670 to the University of Colorado Boulder, the Institute of Behavioral Science, Earth Lab, the Cooperative Institute for Research in Environmental Sciences, the Grand Challenge Initiative and the Innovative Seed Grant program at the
University of Colorado Boulder as well as the Eunice Kennedy Shriver National Institute of Child Health & Human Development of the National Institutes of Health under Award Numbers R21 HD098717 01A1 and P2CHD066613. The content is solely the responsibility of the authors and does not necessarily represent the official views of the National Institutes of Health. We gratefully acknowledge access to the Zillow Transaction and Assessment Dataset (ZTRAX) through a data use agreement between the University of Colorado Boulder and Zillow Group, Inc. More information on accessing the data can be found at http://www.zillow.com/ztrax. The results and opinions are those of the
authors and do not reflect the position of Zillow Group. Support by Zillow Group, Inc. is gratefully acknowledged.

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

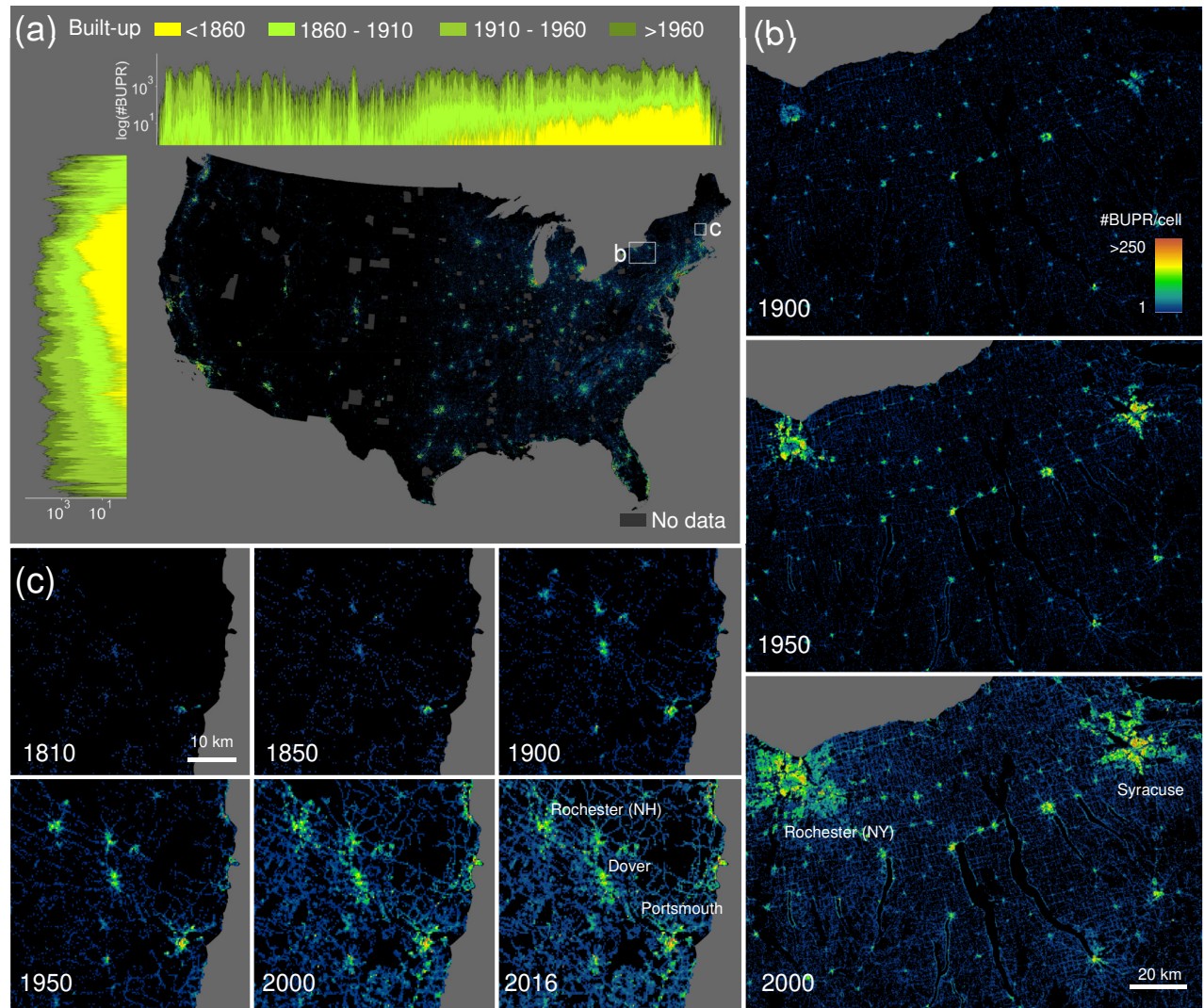

**Figure 1.** Fine-resolution time series of gridded building data for the U.S.: (a) contemporary (2016) built-up property records (BUPR) in the US, including log-transformed directional (i.e., North-South and East-West) histograms for different time periods; also shown are counties of missing data, (b) BUPR time series in mixed urban-rural context shown for the Syracuse – Rochester region (New York) for 1900, 1950, and 2016, and (c) long-term BUPR time series covering the whole time period 1810 – 2016 showing early settlements in New Hampshire and their development patterns.

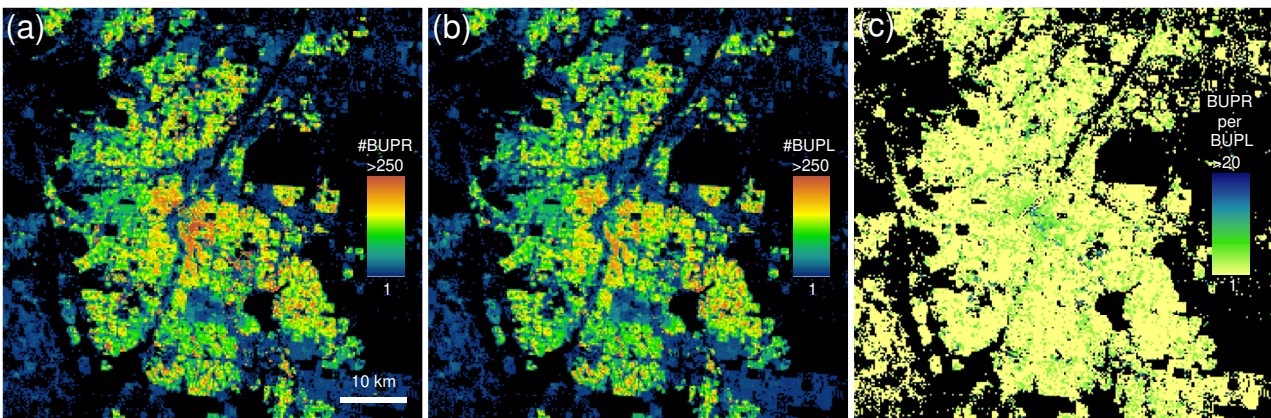

**Figure 2.** Comparison of (a) built-up property records, and (b) built-up property location surfaces, shown for Denver, Colorado; (c) cell-by-cell ratio surface (i.e., built-up property records per built-up property location) highlighting the presence of structures of multi-address or multi-owner records, representing large / high-rise office or apartment buildings, or condominiums.

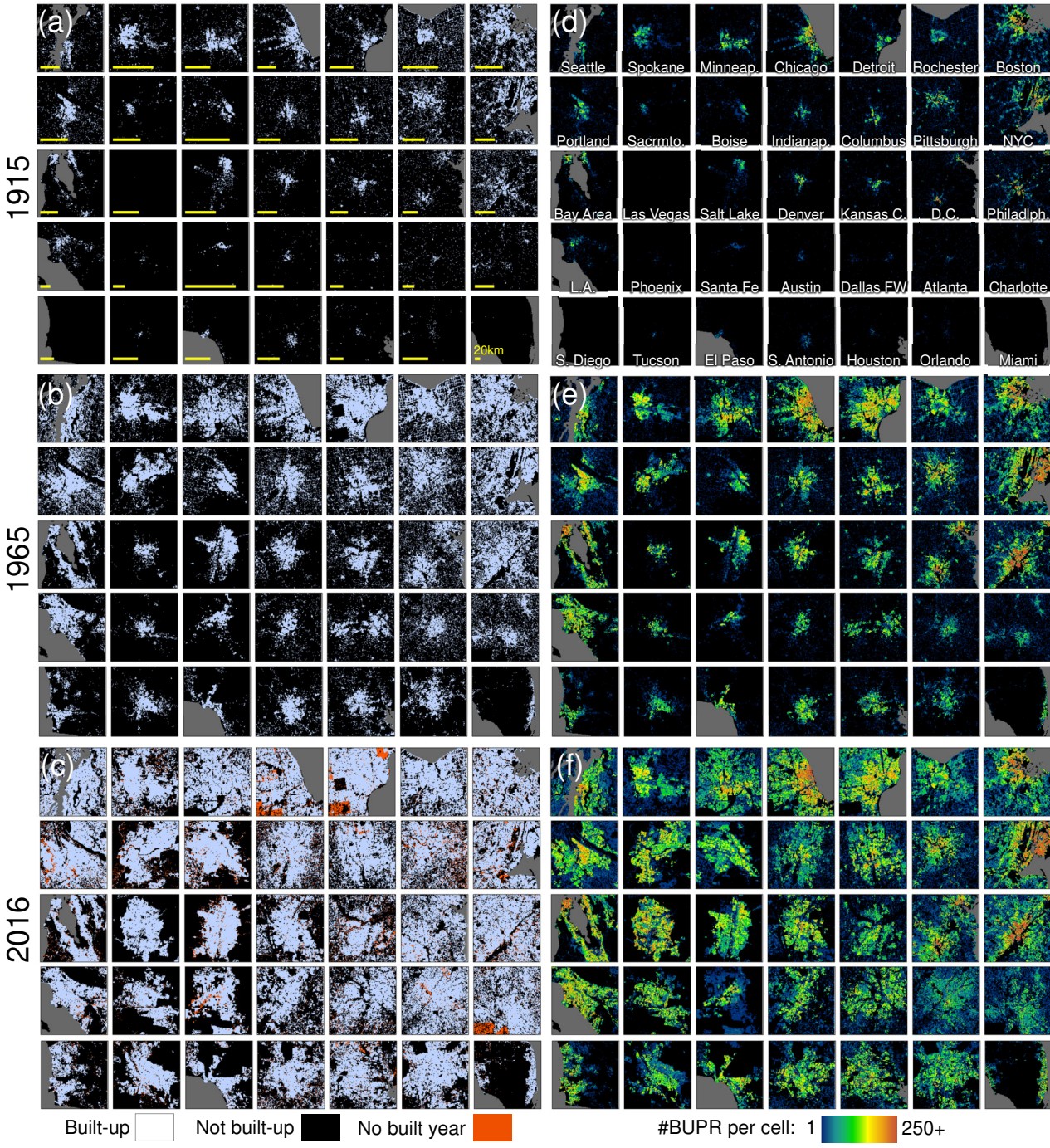

**Figure 3.** Built-up area (BUA) surfaces for 35 selected U.S. cities in (a) 1915, (b) 1965, and (c) 2016, and (d-f) corresponding BUPR surfaces. Cities are arranged in a quasi-geographic space, i.e., Northeastern cities shown in the upper right part of the panels, etc. Panel (c) also shows grid cells where no built year information is available.Note that cities are depicted at individual scales; see 20 km scale bars in panel (a) and Fig. A1 for size relationships between shown extents.

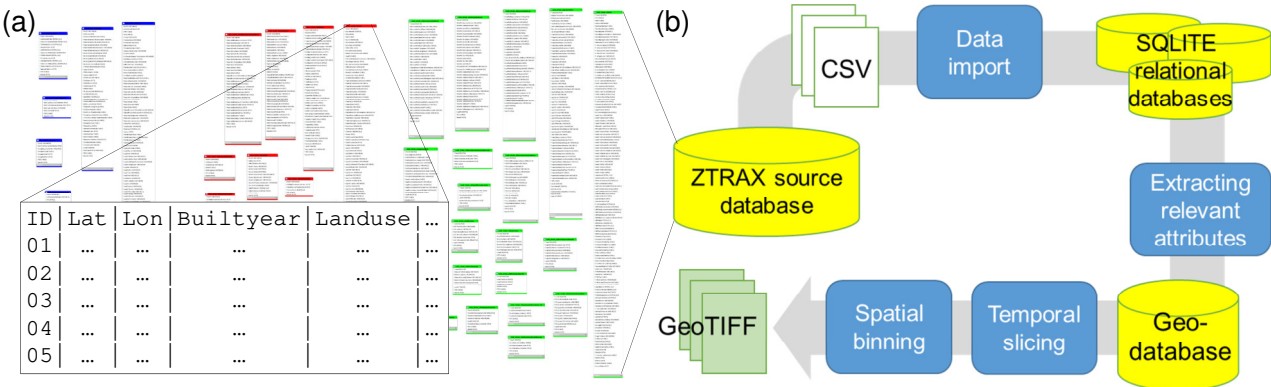

**Figure 4.** (a) Entity diagram illustrating the complexity of the ZTRAX data model, showing each database table and table attributes, and (b) generalized processing workflow to generate the BUPR, BUPL, and BUA surface series based on ZTRAX data records.

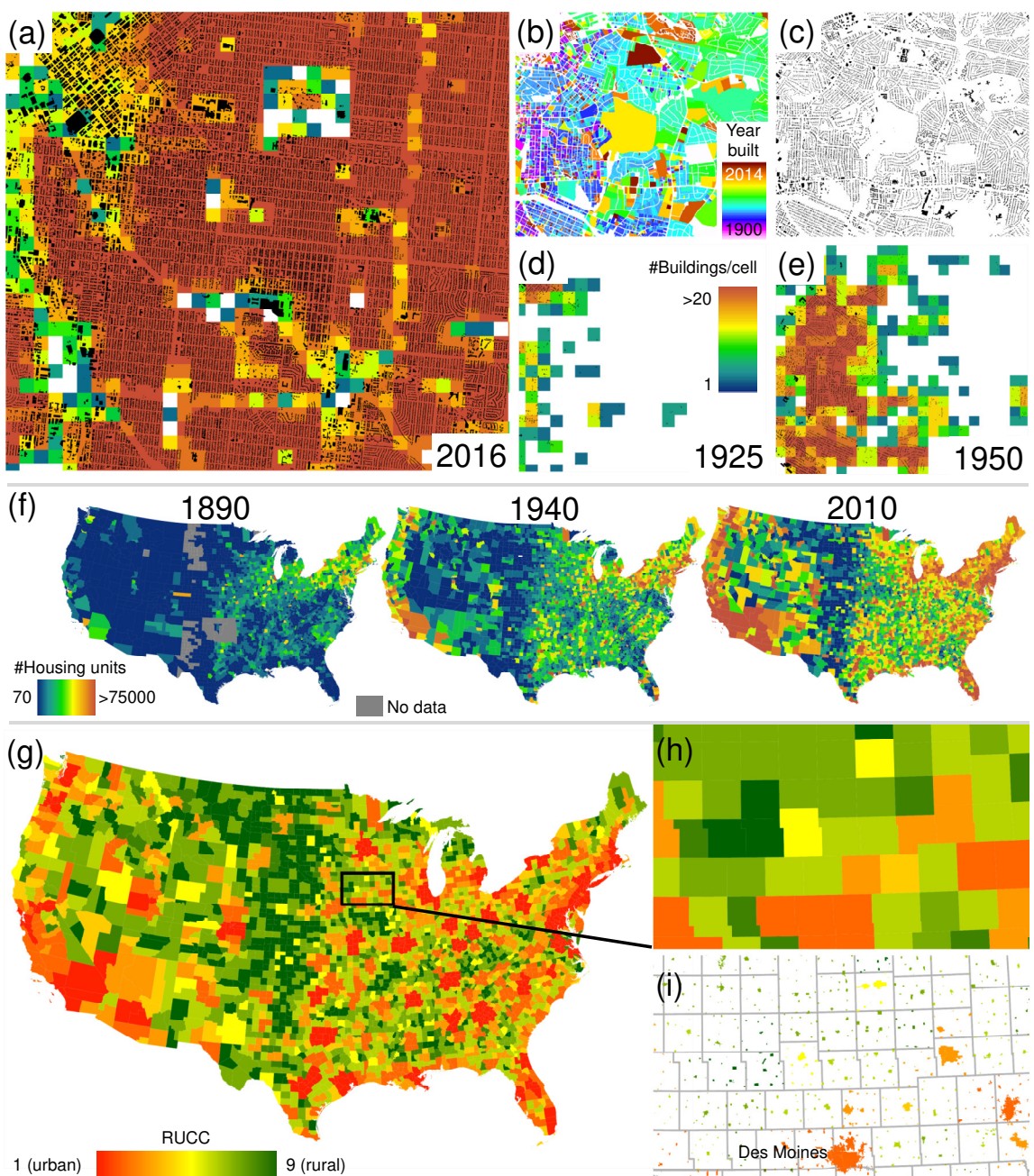

**Figure 5.** Datasets used for validation of the created surfaces: (a) contemporary US-wide building count surface, generated from the Microsoft building footprint data (overlaid) by aggregating to grid cells of 250 m spatial resolution, shown for downtown Denver, Colorado, (b) and (c): multi-temporal building footprint data available for 30 counties in the US, shown for a region in Charlotte, North Carolina, (d) and (e) resulting building count surfaces for 1925, and 1950, respectively, and (f) U.S. Census based dwelling statistics for U.S. counties in 1890, 1940, and 2010. (g) 2013 county-level USDA rural-urban continuum codes (RUCC), (h) enlargement of the county-level RUCC data around Des Moines (Iowa), and (i) RUC codes attached to U.S. census-designated places in 2010 for the same area.

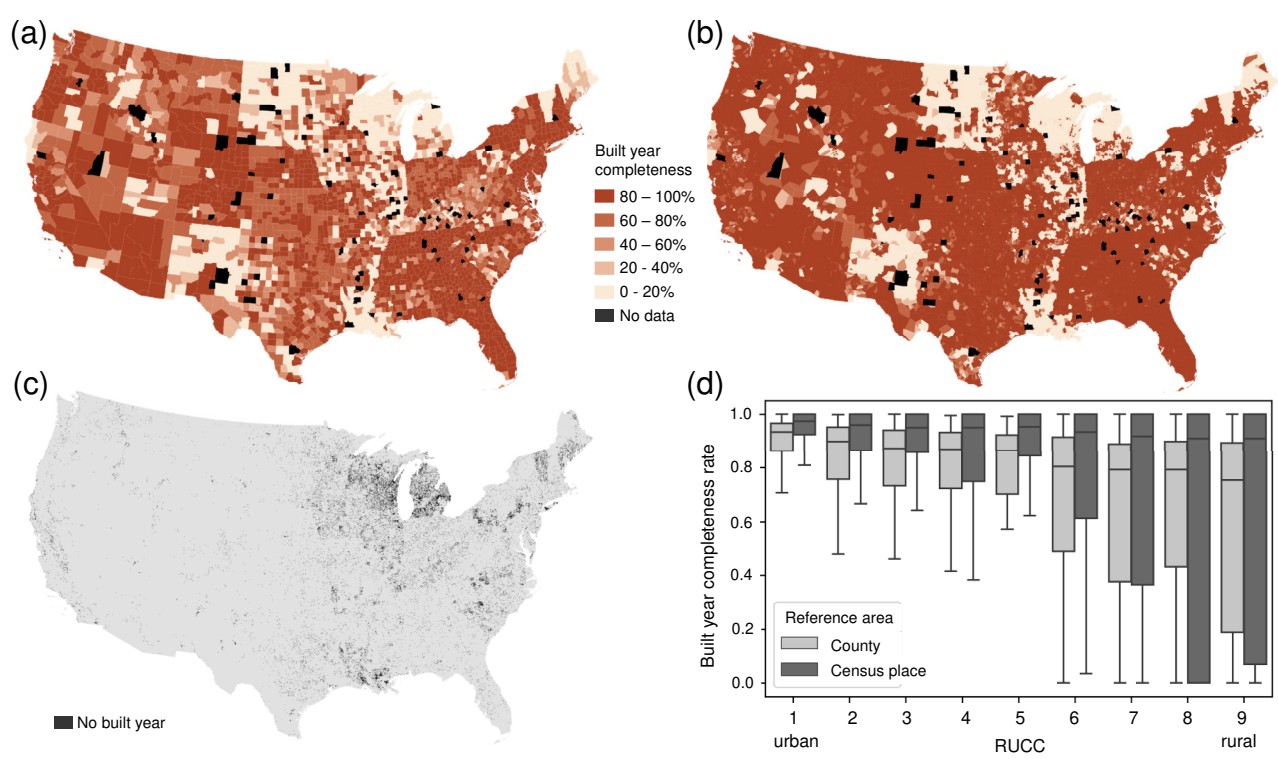

**Figure 6.** Data completeness analysis. (a) Built-year county-level completeness, and (b) census place level completeness, (c) grid cells without built year information, and (d) trends of built-year completeness across the rural-urban continuum. Census place boundaries shown in (b) are generalized using Thiessen polygons derived from place polygon centroids for readability purposes.

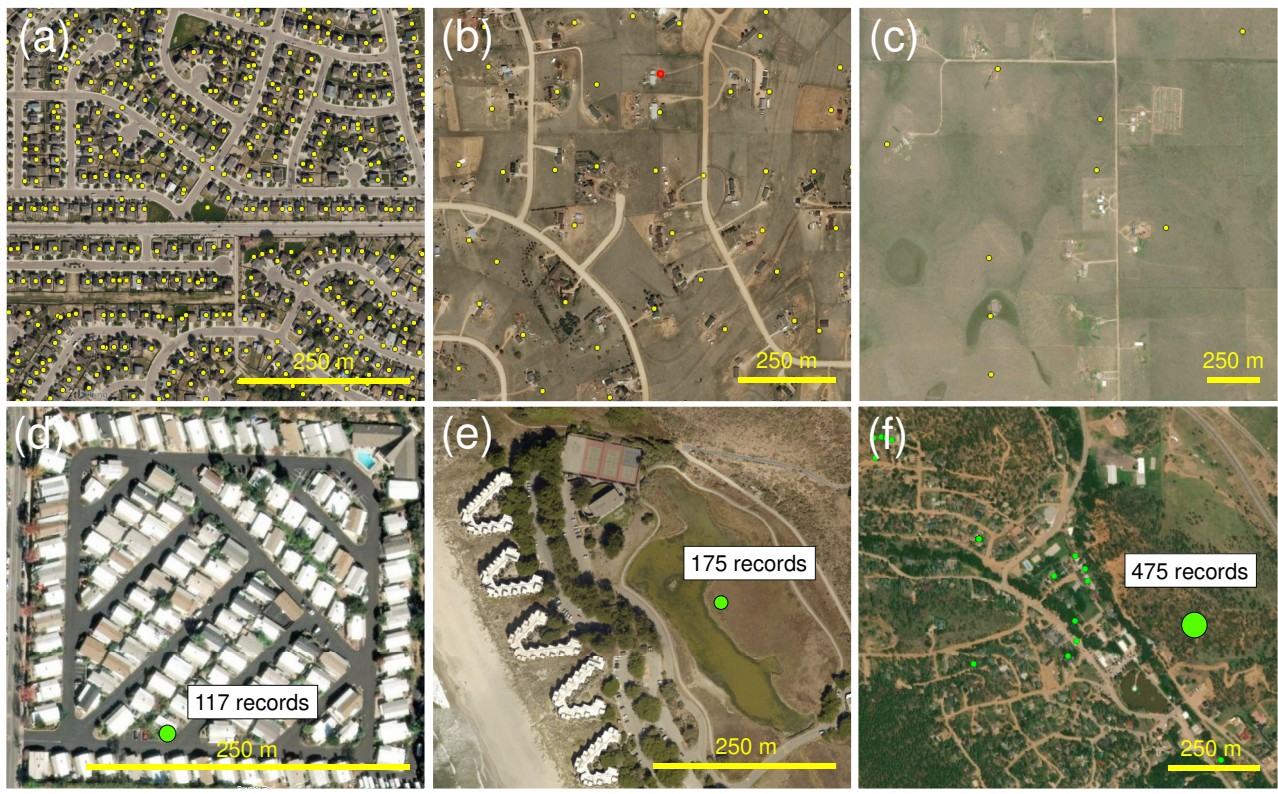

**Figure 7.** Variations of positional accuracy and generalization levels in the ZTRAX database: Example of (a) highly accurate settlement locations in a dense residential neighbourhood dominated by single-family homes, (b) settlement locations of medium positional accuracy, and (c) of low positional accuracy in rural parts of the U.S. Spatially generalized settlement locations (i.e., multi-record locations) for (d) a mobile home park, and (e) a planned community / condominium; (f) a rare agglomeration of records, likely resulting from pseudo-locations assigned during database work in progress. Base map imagery from © Microsoft.

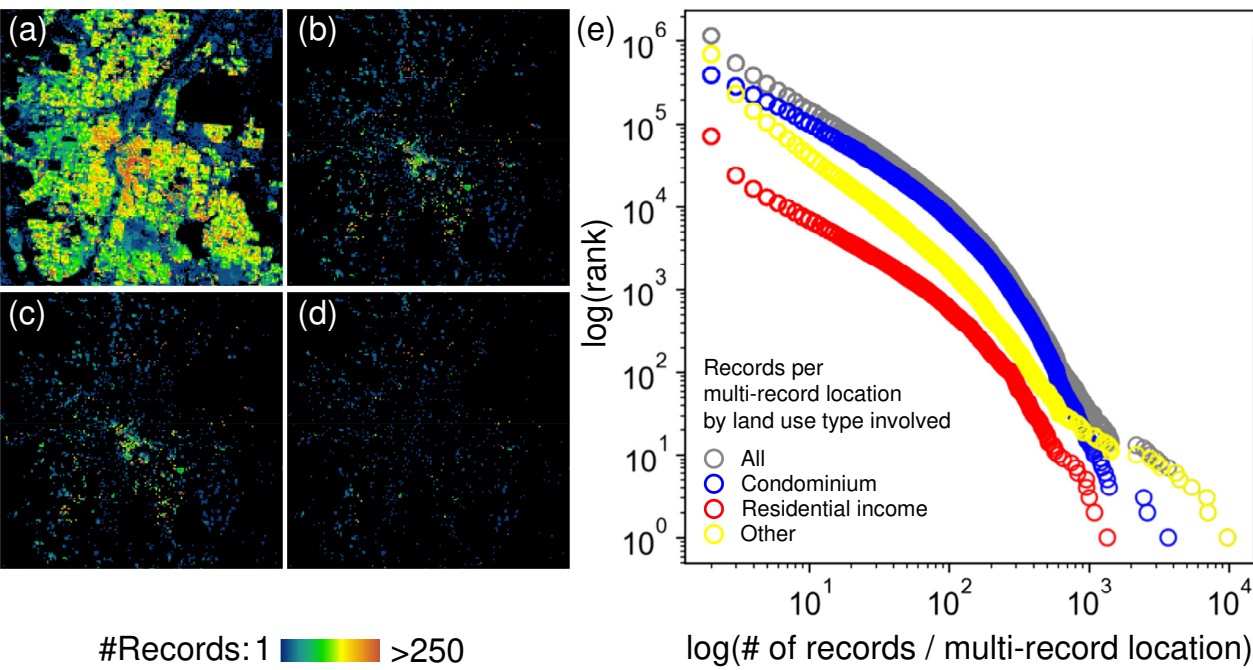

**Figure 8.** Analysis of multi-record locations. (a) BUPR surface for Denver, Colorado, (b) BUPR for multi-records only, (c) BUPR for multi-records without residential income land use, (d) without residential income or condominiums, (e) rank-size plots of multi-record locations (size = number of multi-records per location) for different land use categories.

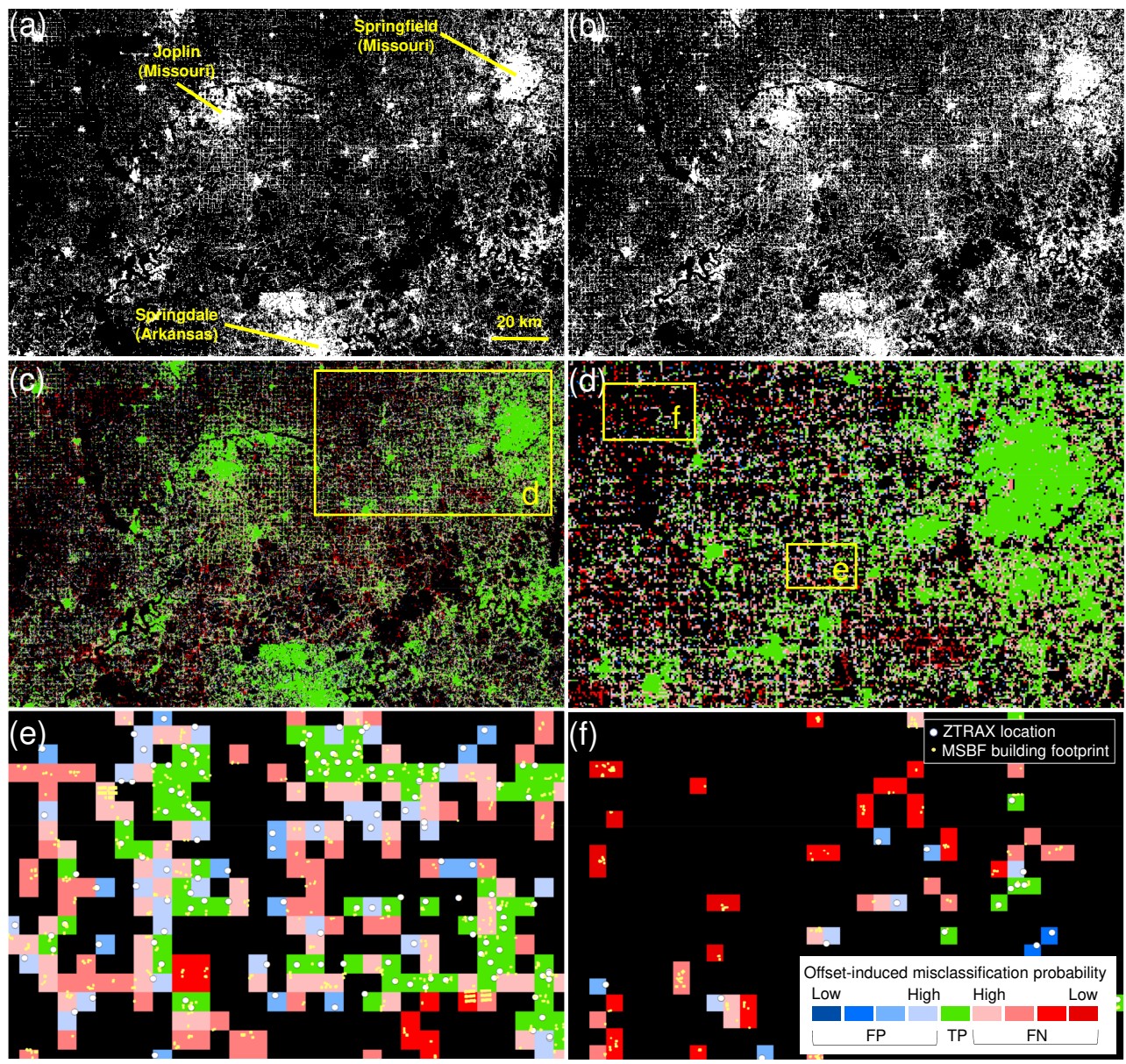

**Figure 9.** Cross-scale positional uncertainty surfaces: (a) "Contemporary", ZTRAX-derived settled areas (i.e., BUA surface from 2016), (b) corresponding reference surface derived from MSBF data, (c) resulting spatial disagreement surface indicating the estimated offset-induced misclassification probability, (d) subset shown for a region west of Springfield, Missouri, and enlargements showing regions characterized by (e) disagreement likely introduced by spatial offsets, and (f) false negatives unlikely introduced by spatial offsets, but rather by missing data.

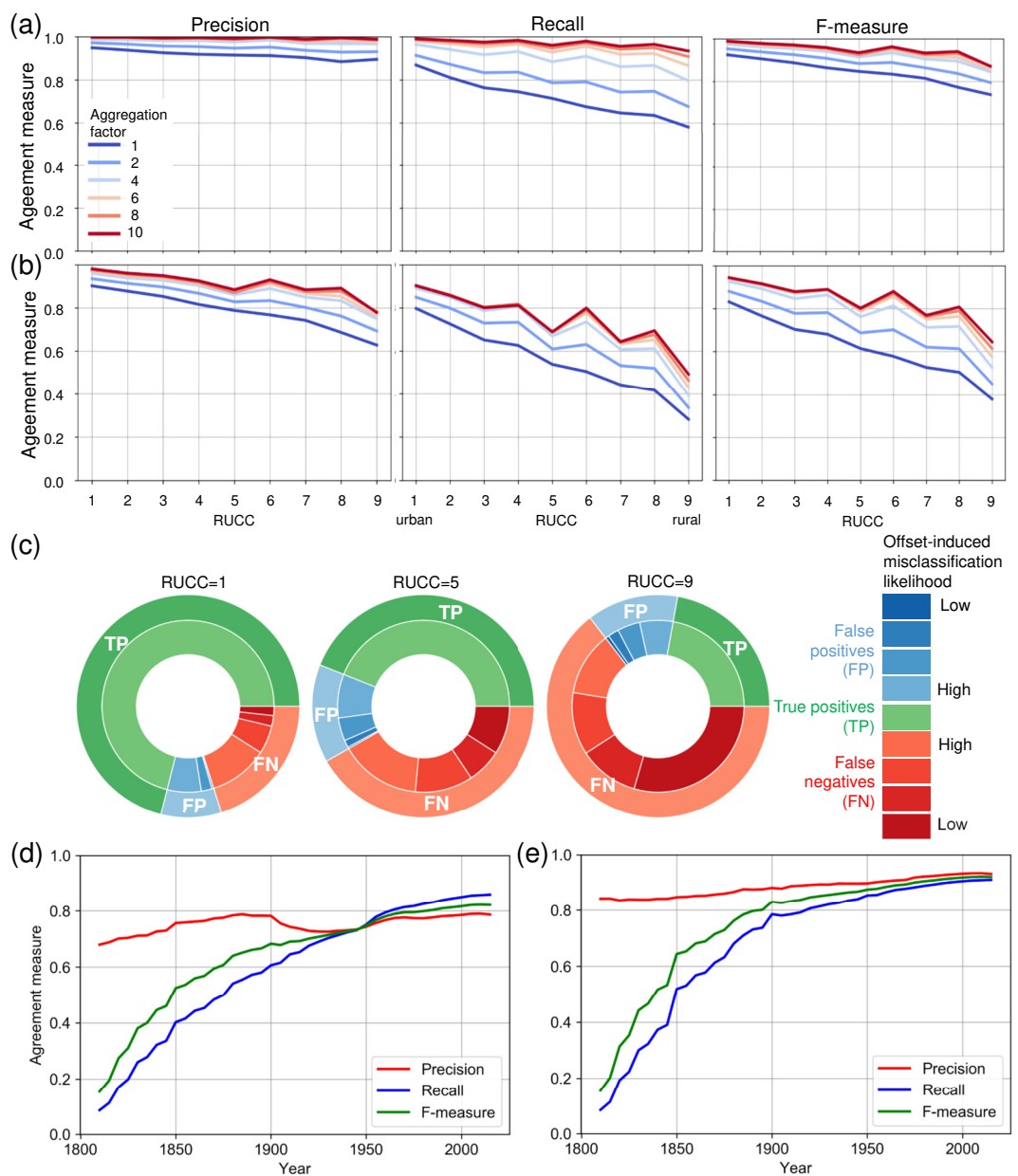

**Figure 10.** Positional accuracy assessment results: (a) Precision, recall, and F-measure between contemporary built-up grid cells derived from the 2016 BUPR surface across the rural-urban continuum and for multiple spatial aggregation levels, evaluated within 2010 census place boundaries, and (b) evaluated within all CONUS landmass (excluding 82 counties where no ZTRAX data is available), (c) Pie charts showing the proportions of agreement classes (outer rings) and probability categories of disagreement induced by spatial offsets between test and contemporary building footprint data within each disagreement class (inner rings), shown for strata of RUCCs 1 (highly urban), 5 (intermediate), and 9 (most rural), respectively, and trajectories of accuracy measures over time for (d) low built-up density, and (e) high built-up density counties, against the MTBF30 validation database.

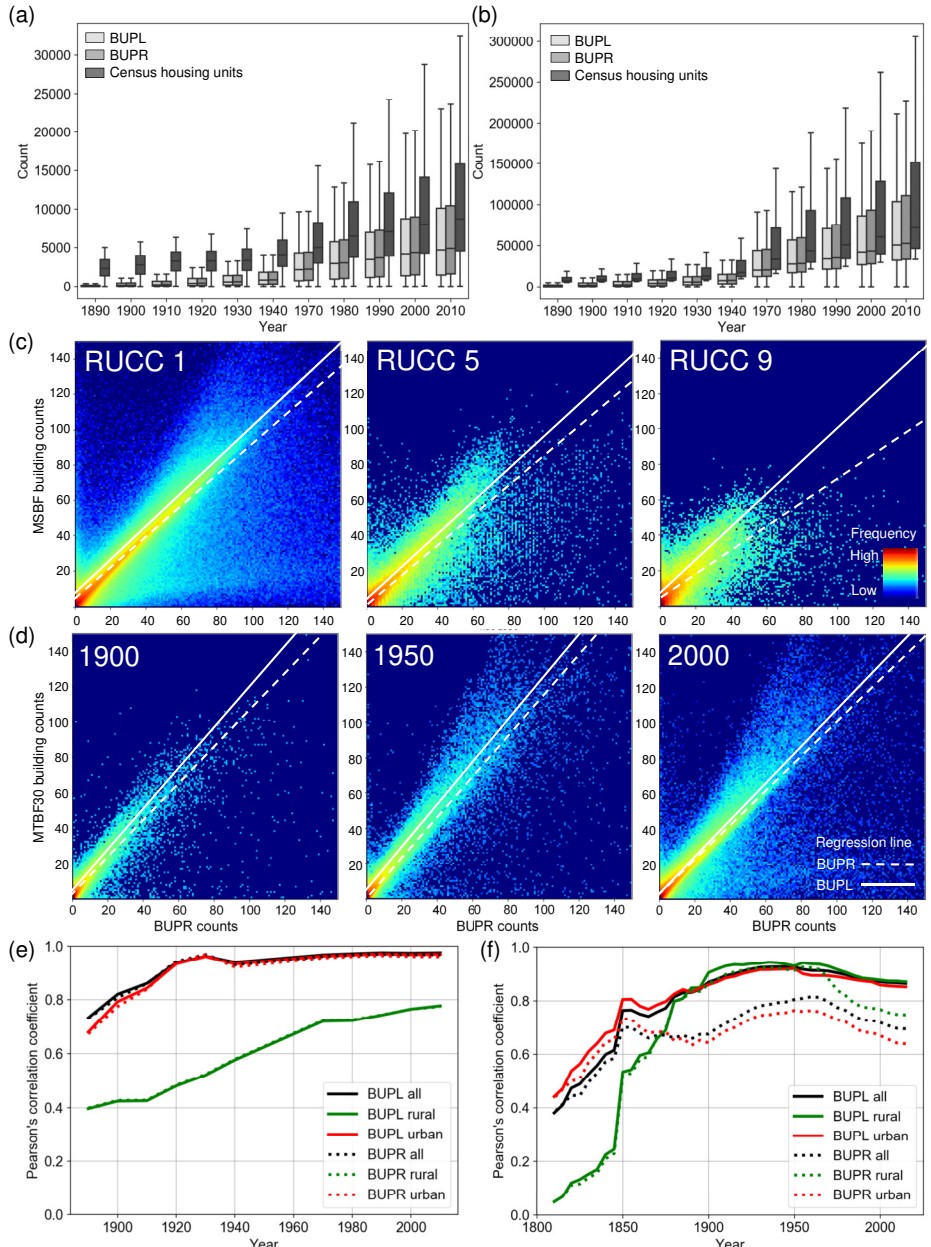

**Figure 11.** Results of the quantity agreement analysis: US-wide trends of housing development from 1890 to 2010 according to U.S. census data and BUPR derived trajectories for strata of (a) rural and (b) urban counties (separated by the 75th percentile of the decennial census data distributions), (c) grid-cell wise quantity agreement between test data and MSBF data in 2016, shown for counties of USDA RUC codes 1 (urban), 5 (intermediate), and 9 (rural), (d) multi-temporal trends of quantity agreement with building counts derived from the MTBF30 database in 1900, 1950, and 2000, and time series of Pearson's correlation coefficients (e) for county-level BUPR / BUPL summaries against U.S. census housing unit counts, and (f) against the multi-temporal building footprint reference database at the 250 m grid cell level. (c) and (d) also show a regression line obtained from robust linear regression.

**Table 1.** Overview of the datasets used for validation of the BUPR, BUPL, and BUA surface series.

| Validation dataset | Measure | Geographic coverage | Temporal coverage | Spatial granularity | Temporal granularity |
|---|---|---|---|---|---|
| Microsoft U.S. building footprint data (MSBF) | Physical built-up structures | CONUS | approx. 2016 (uni-temporal) | building outline | - |
| Multi-temporal building footprint database (MTBF30) | Physical built-up structures | 30 counties in CONUS (Appendix Table B1) | approx. 1800 to 2015 | building outline | annual |
| U.S. Census housing unit counts | Housing units / households | CONUS | 1890 - 2010 | county | between 10 and 30 years |

**Table 2.** Overview of all data products currently contained in the HISDAC-US data compilation.

| Data product | DOI | Data citation |
|---|---|---|
| Built-up property records (BUPR) | https://doi.org/10.7910/DVN/YSWMDR | Uhl, Johannes H.; Leyk, Stefan, 2020, "Historical built-up property records (BUPR) - gridded surfaces for the U.S. from 1810 to 2015", https://doi.org/10.7910/DVN/YSWMDR, Harvard Dataverse, V1 |
| Built-up property locations (BUPL) | https://doi.org/10.7910/DVN/SJ213V | Uhl, Johannes H.; Leyk, Stefan, 2020, "Historical built-up property locations (BUPL) - gridded surfaces for the U.S. from 1810 to 2015", https://doi.org/10.7910/DVN/SJ213V,HarvardDataverse, V1 |
| Built-up areas (BUA) | https://doi.org/10.7910/DVN/J6CYUJ | Uhl, Johannes H.; Leyk, Stefan, 2020, "Historical built-up areas (BUA) - gridded surfaces for the U.S. from 1810 to 2015", https://doi.org/10.7910/DVN/J6CYUJ, Harvard Dataverse, V1 |
| BUPR / BUPL / BUA uncertainty surfaces | https://doi.org/10.7910/DVN/T8H5KF | Uhl, Johannes H.; Leyk, Stefan, 2020, "Uncertainty surfaces accompanying the BUPR, BUPL, and BUA gridded surface series", https://doi.org/10.7910/DVN/T8H5KF, Harvard Dataverse, V1 |
| Published in Leyk and Uhl (2018a): | | |
| Built-up intensity (BUI) | https://doi.org/10.7910/DVN/1WB9E4 | Leyk, Stefan; Uhl, Johannes H., 2018, "Historical built-up intensity layer series for the U.S. 1810 - 2015", https://doi.org/10.7910/DVN/1WB9E4, Harvard Dataverse, V2 |
| First built-up year composite (FBUY) | https://doi.org/10.7910/DVN/PKJ90M | Leyk, Stefan; Uhl, Johannes H., 2018, "Historical settlement composite layer for the U.S. 1810 - 2015",https://doi.org/10.7910/DVN/PKJ90M, Harvard Dataverse, V2 |
| County-level uncertainty statistics | https://doi.org/10.7910/DVN/CXD9BW | Leyk, Stefan; Uhl, Johannes H., 2018, "County-level uncertainty statistics accompanying the historical settlement layers for the U.S. 1810 - 2015", https://doi.org/10.7910/DVN/CXD9BW, Harvard Dataverse, V2 |

## Appendix A: Geographic coverage of the MTBF30 building footprint database.

**Table A1.** List of 30 counties covered by the multi-temporal building footprint (MTBF30) database.

| County | State | Population 2015 | Area [$km^2$] | % built-up according to reference data |
|---|---|---|---|---|
| **Low-density counties (stratification in 2015)** | | | | |
| Benton County | Oregon | 86414 | 1747 | 0.9 |
| Franklin County | Massachusetts | 70927 | 1876.8 | 1.9 |
| Berkshire County | Massachusetts | 128565 | 2451 | 2.6 |
| Boulder County | Colorado | 313864 | 1780.4 | 3.3 |
| Hampshire County | Massachusetts | 161106 | 1413.5 | 4.1 |
| Carver County | Minnesota | 97396 | 970 | 4.2 |
| Dukes County | Massachusetts | 17320 | 319.5 | 4.6 |
| Manatee County | Florida | 351771 | 2064.4 | 6.5 |
| Nantucket County | Massachusetts | 10821 | 155.5 | 6.5 |
| Worcester County | Massachusetts | 814972 | 4087.1 | 6.5 |
| Washington County | Minnesota | 249320 | 1092.8 | 7.0 |
| Dakota County | Minnesota | 412182 | 1522.3 | 7.8 |
| Hampden County | Massachusetts | 469566 | 1641.9 | 8.3 |
| Plymouth County | Massachusetts | 507050 | 1822.3 | 9.3 |
| Vanderburgh County | Indiana | 181918 | 609.2 | 9.7 |
| Anoka County | Minnesota | 341742 | 1153.1 | 10.3 |
| Sarasota County | Florida | 397024 | 1569.5 | 11.2 |
| Bristol County | Massachusetts | 554626 | 1529.6 | 11.4 |
| Essex County | Massachusetts | 770486 | 1388 | 12.2 |
| Barnstable County | Massachusetts | 214665 | 1177.8 | 12.3 |
| Baltimore County | Maryland | 827794 | 1623.9 | 12.6 |
| Hillsborough County | Florida | 1318325 | 2800.3 | 13.4 |
| Monmouth County | New Jersey | 629018 | 1255.9 | 15.5 |
| **High-density counties (stratification in 2015)** | | | | |
| Norfolk County | Massachusetts | 692688 | 1083.9 | 16.5 |
| Middlesex County | Massachusetts | 1572523 | 2196.6 | 16.7 |
| Hennepin County | Minnesota | 1212097 | 1566.4 | 20.8 |
| Mecklenburg County | North Carolina | 1011928 | 1409.6 | 23.8 |
| Ramsey County | Minnesota | 533634 | 439.5 | 30.8 |
| Suffolk County | Massachusetts | 769809 | 177.9 | 38.2 |
| New York City | New York | 8537673 | 781.1 | 54.3 |

**Appendix B:  City extent comparison.**

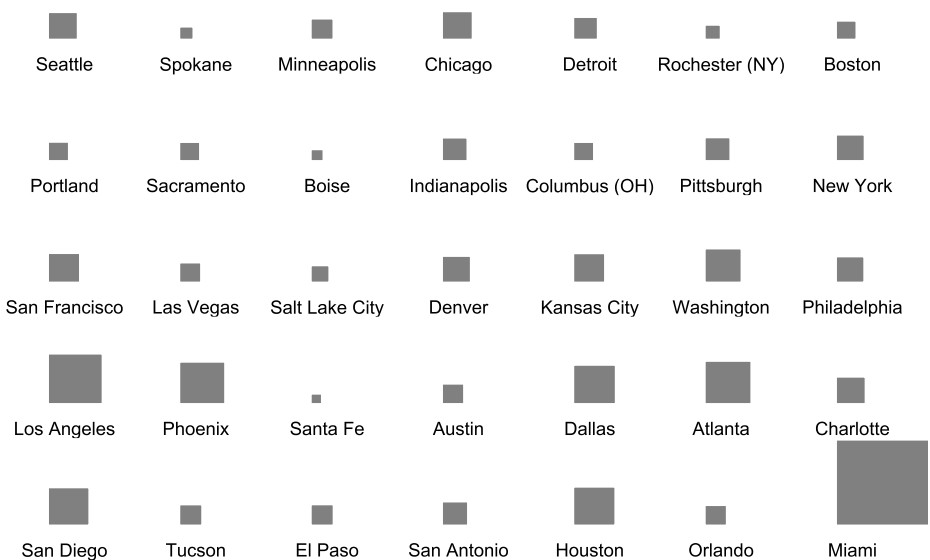

**Figure B1.** Size relationships between city extents shown in Fig. 3.

## Appendix C:  Assessing the effects of public building / housing omission

Based on the auxiliary datasets described in Section 3.2.5, we calculated county-level sums of public structures from the USGS national structures dataset (e.g., schools, hospitals, governmental buildings), and of publicly-owned housing units (e.g., established for low-income renters by housing assistance programs), reported by the HUD, and covering 1,934 counties in the conterminous U.S. Moreover, we calculated the number of public amenities, reported in Open Street Map, as a cross-comparison to the public structures reported by the USGS. More specifically, we used objects from the OSM database with the

key "amenity" that are tagged as one of the following usage types: public building, townhall, library, police, hospital, school, community center, university, social facility, nursing home, clinic, courthouse, monastery, place of worship, post office, prison, or college.

    To quantify the proportion of structures that may be omitted by ZTRAX, we calculated the proportions of these counts with respect to the estimated total number of structures / housing units per county (i.e., the sums of public entities and ZTRAX-

derived counts). As can be seen in Figure C1, these proportions are below 5% for the large majority of counties. Thus, the omission of public properties in ZTRAX causes an underestimation of approximately 5% of the total number of BUPR / BUPL in most counties. For detailed analyses at local scales, users may employ the described auxiliary datasets (Section 3.2.5) to mitigate these omission errors.

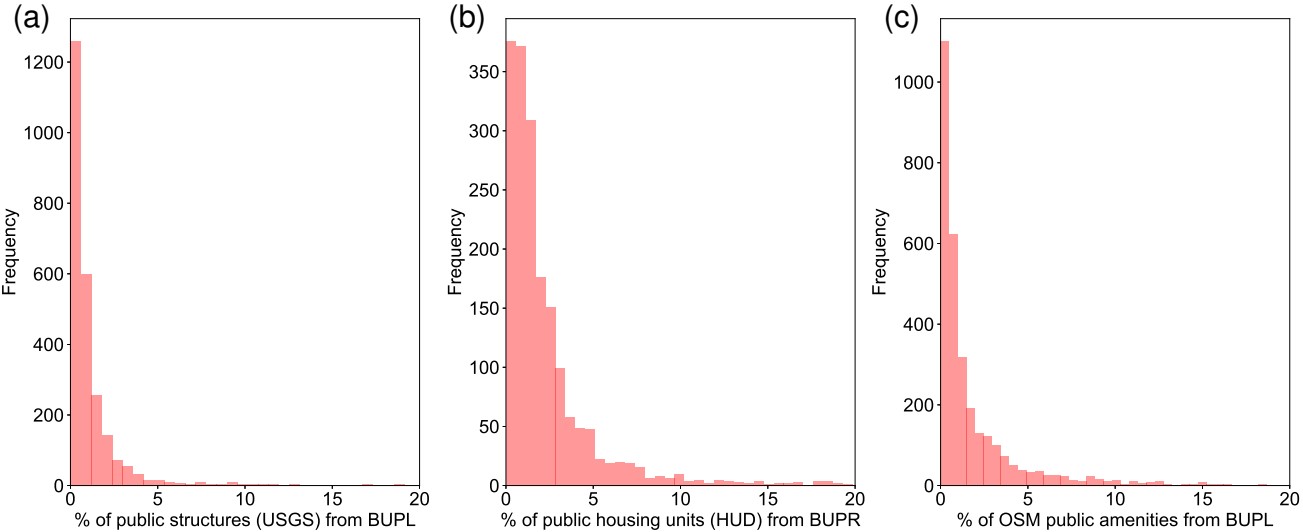

**Figure C1.** Assessment of omission errors introduced by lacking information on publicly-owned buildings in ZTRAX: Frequencies of county-level proportions of (a) public structures, (b) public housing units, and (c) public amenities, referred to the respective county-level sums of BUPR or BUPL. Note that Y-axis range differs by panel.

**Appendix D: Assessment of multi-record built-up property locations across different domains.**

We analysed the usage types of multi-record locations at the county level across different domains. Counties with high numbers of multi-records (Fig. D1a) and counties with high built-up density (Fig. D1b) exhibit high proportions of office / residential condominiums. Moreover, the total building indoor area reported at multi-record locations is greater when condominiums are involved (Fig. D1c). Conversely, we observe narrow built year ranges at multi-record locations involving condominiums (Fig. D1d). These trends reflect some general characteristics of condominiums and planned communities: They tend to be built-up

in short periods of time, rather close to densely than sparsely populated regions, and constitute large shares of the local built-up intensity. Analyzing the distributions of multi-records for each individual multi-record location, rather than looking at general trends of multi-record locations at the county-level, we see a different picture. As Fig. D1e indicates, large proportions of multi-records are of residential income usage. Also, Fig. D1e suggests that condominium multi-record locations typically have <200 records. Multi-record locations holding larger numbers of records than 300 are less frequent (cf. Fig. 8e), and their usage

patterns are mixed. The yellow bars to the very right in Fig. D1e likely represent the previously described pseudo-locations, i.e., artificial multi-record locations not representing residential income or condominiums. As can be seen, these cases represent only a minor proportion of all multi-record locations in the U.S. and can be masked out or subtracted using the uncertainty surface provided (Sect. 4.4.1).

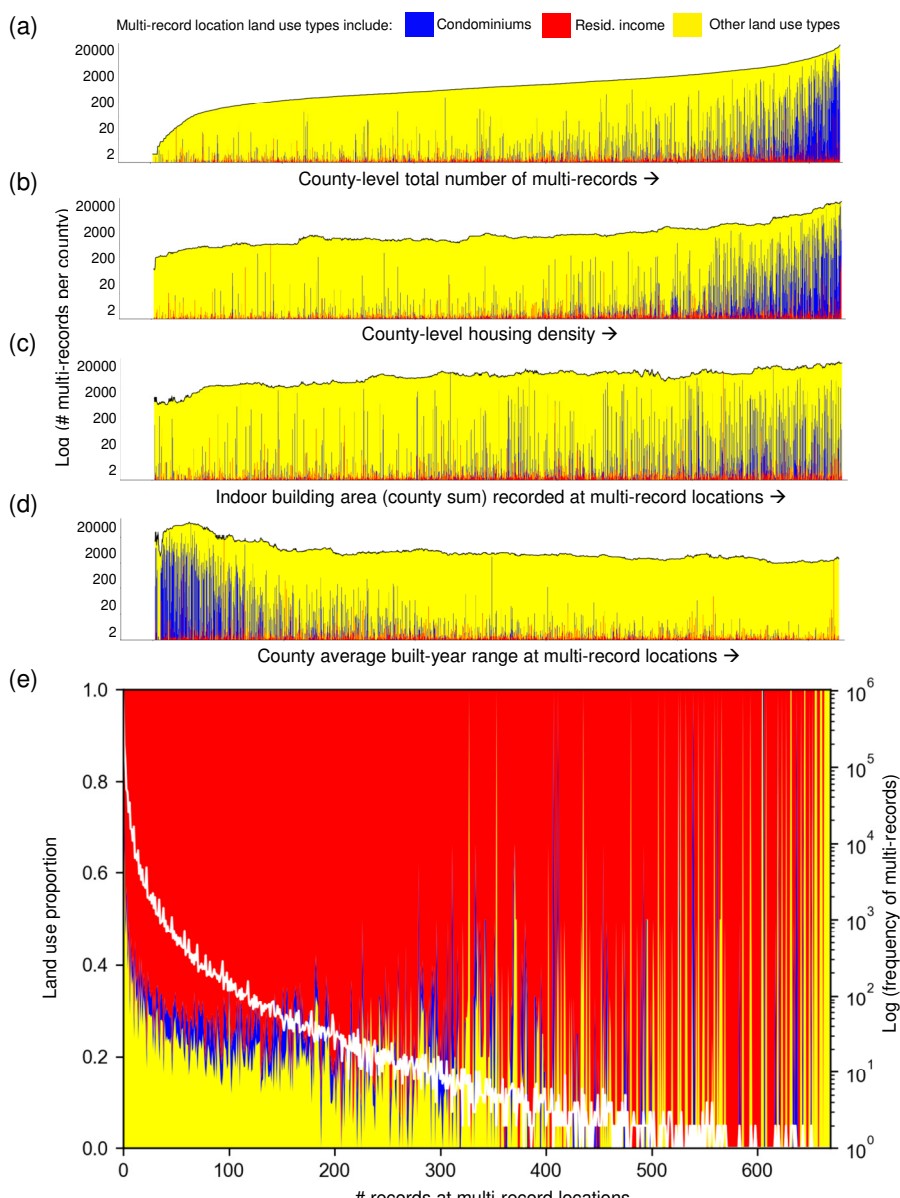

**Figure D1.** Detailed analysis of multi-record locations across different domains: Stacked bar plots with each bar representing the proportions of involved land use types at multi-record locations per county. Lengths of the bars represent the log-transformed total number of multi-records per county, and the horizontal sorting of the bars from left to right is based on (a) the number of multi-records per county, (b) county-level housing density derived from 2010 U.S. census data (see Sect. 3.2.3), (c) the sum of indoor building area over all multi-records per county, and (d) the built-year range recorded at multi-record locations (county averages). (e) Distributions of the number of records at multi-record locations and their land use proportions, overlaid with the log-transformed total number of multi-records (white). Sorted data series in (a) – (d) were smoothened by a sliding median filter (size = 50) for improved readability.

## Appendix E: Multi-scale accuracy assessment and offset-induced misclassification probability modelling.

First, classical map comparison is conducted for the ”contemporary” $BUA_{2016}$ surface (Fig. 9a) and reference surface (Fig. 9b) at original spatial resolution (here: 250 m), resulting in a categorical gridded surface indicating the agreement type (true positives, true negatives, false positives, false negatives, i.e., $TP_{250}$, $TN_{250}$, $FP_{250}$, $FN_{250}$, respectively). Then, both $BUA_{2016}$ and the reference surfaces are downsampled by factor 2, and agreement types per grid cell are re-computed (i.e., $TP_{500}$, $TN_{500}$, $FP_{500}$, $FN_{500}$). This is done iteratively for a specified number of downsampling factors (here: up to factor 4, corresponding to a cell size of 2000 m), which indicates the spatial range within which offsets as described above are assumed to occur. The agreement type surfaces of all downsampled levels are then upsampled to the native resolution (i.e., 250 m) and stacked into a multi-scale data cube (Fig. E1). Based on this cube, cross-scale trajectories per grid cell are extracted for each grid cell that was misclassified at native resolution (Table E1). When a cross-scale trajectory switches from FP to TP, or from FN to TP, respectively, a ***probability of offset-induced misclassification*** is assigned to the grid cell as a function of the aggregation level where this switch occurs. This probability is lowest for grid cells that remain in FP or FN categories across all scales, and highest if the switch to TP occurs immediately after the first downsampling step. Subsets of resulting surfaces indicating FPs and FNs including their estimated offset-induced misclassification probability, as well as the TPs, are shown in Fig. 9c-f.

**Table E1.** Cross-scale disagreement trajectories and assigned offset-induced misclassification probability.

| Spatial aggregation level | | | | Probability of offset-induced misclassification |
|---|---|---|---|---|
| 250 m | 500m | 1000m | 2000m | |
| FP | FP | FP | FP | lowest |
| FP | FP | FP | TP | low |
| FP | FP | TP | TP | medium |
| FP | TP | TP | TP | highest |
| FN | FN | FN | FN | lowest |
| FN | FN | FN | TP | low |
| FN | FN | TP | TP | medium |
| FN | TP | TP | TP | highest |

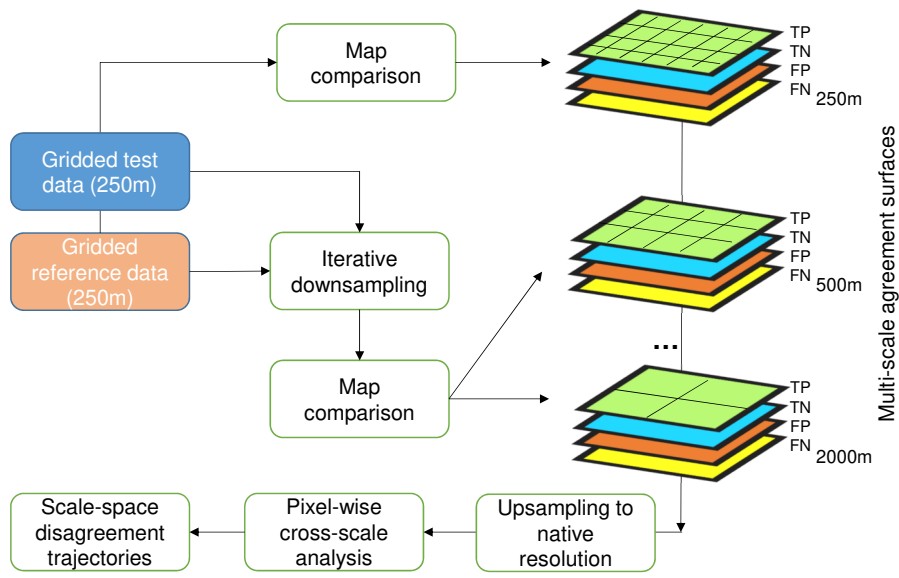

**Figure E1.** Processing workflow to generate the cross-scale disagreement composite surface.

**Appendix F:  Cross-comparing building footprint validation datasets.**

Linking the accuracies obtained for the most recent point in time of the multi-temporal accuracy assessment (Sect. 4.2.3, Fig.
10d,e) to the U.S.-wide, contemporary results (Sect. 4.2.2), we observe lower recall values when validating against MSBF data
(0.3 in rural, and 0.8 in urban counties, Fig. 10b) as compared to the recall obtained when validating against the MTBF30
data (0.85 in low-density counties, vs. 0.9 in high-density counties). This effect could be due to a sampling bias as a result of
comparing accuracy measures derived across 30 counties, selected on basis of data availability, against approximately 3,000
counties. Another possible cause could be high commission errors (i.e., lower levels of precision) in the MSBF data, for
which, to our knowledge, no thorough validation study has been published. Thus, we evaluated the spatial agreement between
the binary reference surfaces derived from MSBF, approximately representing built-up grid cells in 2016, and the surface
derived from the MTBF30 database in 2015, respectively. Considering the latter surface as reference, we observe remarkably
lower levels of precision in lower-density counties (i.e., 0.854, see Table F1) than the overall measure reported by Microsoft
(precision=0.993; Microsoft, 2018). While we would like to emphasize that the results reported in Table F1 need to be further
evaluated critically, since the validation dataset only covers approx. 1% of U.S. counties, they partially explain the low recall
values for the 2016 BUPR surface reported in Sect. 4.2.2. Thus, it is possible that there is a bias in the MSBF data resulting in
higher than expected commission errors in rural areas.

**Table F1.** Cross-comparison of MSBF against building footprints from integrated 30 counties database.

| Agreement measure | All counties | High-density counties | Low-density counties |
| --- | --- | --- | --- |
| PCC | 0.933 | 0.969 | 0.919 |
| Precision (UA) | 0.901 | 0.990 | 0.854 |
| Recall (PA) | 0.957 | 0.960 | 0.955 |
| F-measure | 0.928 | 0.975 | 0.901 |
| Kappa index | 0.866 | 0.935 | 0.834 |

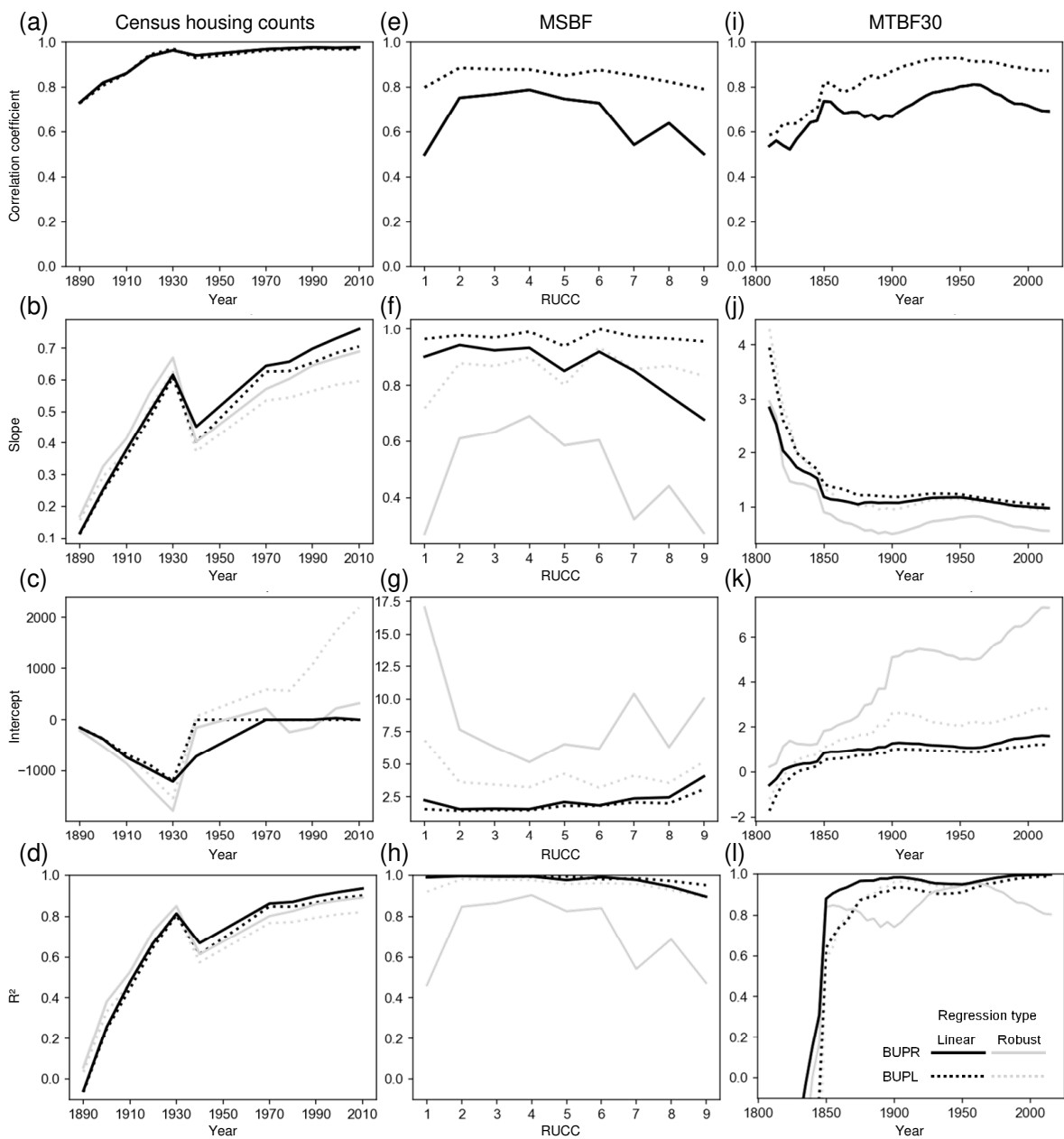

**Figure G1.** Regression and correlation results of BUPR and BUPL counts against the three validation datasets, (a)-(d) against census housing units within historical county boundaries, (e)-(h) against MSBF data across the rural-urban continuum (i.e., 2013 USDA RUC codes at the county level), and (i)-(l) against the MTBF30 database within grid cells. Larger differences between linear and robust regression coefficients indicate the presence of larger numbers of outliers (e.g., planned communities, pseudo-locations).

**Appendix G:  Quantitative agreement assessment of BUPR / BUPL and reference datasets.**

The quantity agreement assessment in Sect. 4.3.3 illustrates the levels of association and correlation between validation datasets
and the BUPR / BUPL surfaces but do not provide quantitative measures of difference. To quantify the differences between
ZTRAX-derived BUPR / BUPL counts ($C_{\text{ZTX}}$) and the counts reported in the three reference datasets ($C_{\text{REF}}$), we define the
absolute count difference $ACD$ and the relative count difference $RCD$ as

$$ACD_{\text{i}} = C_{\text{ZTX,i}} - C_{\text{REF,i}} \tag{G1}$$

and

$$RCD_{\text{i}} = \frac{ACD_{\text{i}}}{C_{\text{REF,i}}} = \frac{(C_{\text{ZTX,i}} - C_{\text{REF,i}})}{C_{\text{REF,i}}} \tag{G2}$$

with $i$ denoting a specific analytical instance or unit (i.e., county or grid cell). The design of these measures will result in neg-
ative values, if the ZTRAX-derived variables underestimate reference counts, and vice versa. We observe several trends. First,
the absolute magnitude of $ACD$ generally increases from rural (low-density) towards urban (high-density) strata (Appendix
Fig. G2a,c,e). Second, magnitudes of $ACD$ to census-derived housing unit counts are lower for BUPR than for BUPL (Ap-
pendix Fig. G2a), confirming our previous observation that BUPR is stronger related to housing units than BUPL. Moreover,
we observe slightly increasing underestimation of MSBF counts towards rural counties (Appendix Fig. G2c), in particular for
the BUPL counts. This trend is even more apparent for the relative measure $RCD$ across RUCC classes (Appendix Fig. G2d).
Interestingly, $ACD$ trends over time in urban counties (Appendix Fig. G2e) show a varying trend across the 19th and 20th
century, exhibiting maximum levels of building count underestimation in the 1950s, particularly visible in the BUPL-derived
$ACD$. The downwards trend prior to 1950 (i.e., increasing underestimation of building counts) could reflect the increasing
establishment of single-family homes, during the primary era of U.S. suburbanization, which are more likely to have addi-
tional, smaller buildings such as sheds or garages, contained in the reference building database. The subsequent upwards trend
post 1950 may be due to increasing building of multi-apartment buildings, condominiums, etc., which mitigates this effect and
results in lower levels of building count underestimation. The relative measure $RCD$ shown in Appendix Fig. G2b and f illus-
trates the count differences with respect to the validation data counts across time, both exhibiting lower magnitudes towards
more recent years, confirming previously made observations of increasing data reliability over time.

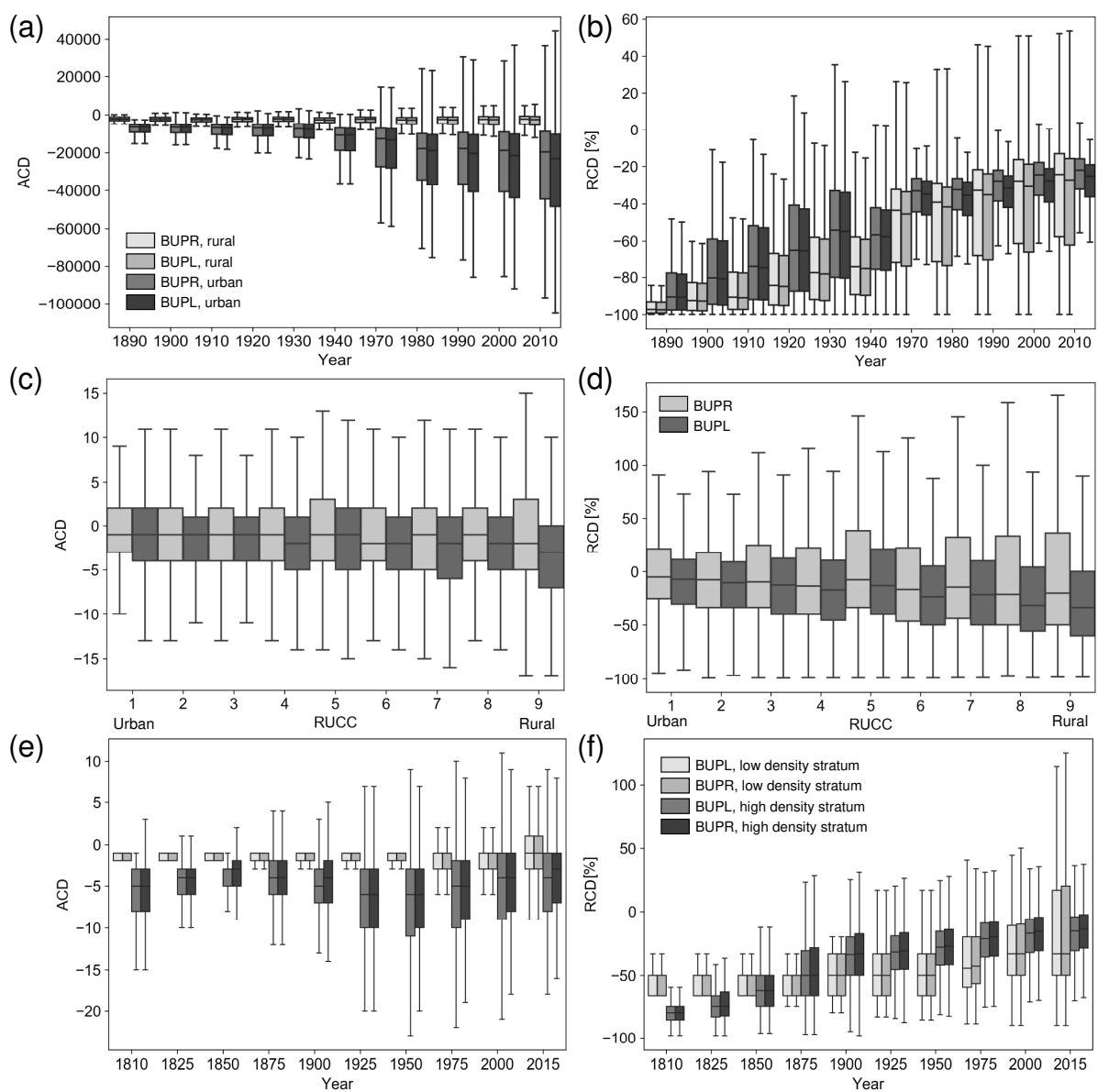

**Figure G2.** Absolute and relative count differences between BUPR, BUPL and validation datasets: (a) absolute differences between census housing unit counts and BUPR / BUPL county aggregates over time, calculated within historical county boundaries, (b) corresponding relative differences, (c) grid-cell level absolute difference distributions against MSBF across the rural-urban continuum, derived from MSBF-based building density deciles (d) corresponding relative difference distributions, using reference building density deciles for stratification, (e) temporal trends of grid-cell level absolute differences against the MTBF-30 database, and (f) corresponding relative difference distributions. Urban-rural stratification in (a), (b), (e), and (f) is based on the 75th percentile of reference count distributions per year. Count difference distributions in (e) and (f) are based on 25-year aggregates to achieve sufficiently large sample size.