# Peer review of "Fine-grained, spatio-temporal datasets measuring 200 years of land development in the United States"

_Earth System Science Data, 2020_

## Referee Comment (RC1) · Tracy Kugler (Referee) · 1 Oct 2020

_General Comments_ The data described in this manuscript are an extension of the existing Historical Settlement Data Compilation for the United States (HISDAC-US) data collection. The new data layers include 250m resolution gridded time series (every 5 years from 1890-2015) of the number of built-up records (BUPR), the number of distinct built-up locations (BUPL), and a binary layer indicating the presence of any built-up records (BUA) in the cell. The BUPR and BUPL layers represent new ways of compiling information from the underlying ZTRAX data, while the BUA layer appears to be a refinement/revision of a previously published BUA layer (cf. Leyk et al. 2020 cited

in this manuscript). All three new layers describe dimensions not previously available in the data collection, and I expect they will prove valuable in a variety of research applications investigating the trajectory of the built environment in the U.S.

The data and methods are described clearly and in sufficient detail. In addition, the authors clearly describe sources and implications of potential uncertainties in the data, as well as a thorough series of validation procedures. The data are easily accessible via the Harvard Dataverse, as described in the manuscript, and include useful meta-data. The layer files accompanying the data are particularly useful for visualizing the data.

_Specific Comments_ I have a few questions about the underlying ZTRAX data that would further clarify the development of the data: 1) Lines 139-140 (p. 5) state that the ZTRAX database contains more than 400 million data records, out of which around 150 million contain spatial information. What do the remaining 250 million data records (without spatial information) represent? In other words, what is missing from the final data by not including those records? 2) More generally, what is the universe of the ZTRAX database? Specifically, what, if any, information does it include for structures that were present historically but not in 2016? The conclusion alludes to the "absence of information on building teardowns or replacements" (lines 461-2, p. 15), but I don't believe this absence of information is mentioned earlier. Is the absence complete, or are there some instances where information about non-contemporary structures is present? This should be clarified in the description of the source data in section 3.1. 3) Were lat/long coordinates present in the ZTRAX database for the 150 million records with spatial information? Or did the authors conduct geocoding based on addresses in the ZTRAX database? The manuscript seems to imply the former, but it would help if it were explicitly stated. If I am mis-reading and it is the latter, information about the overall quality of the geocoding should be provided. For example, what proportion of records were successfully geocoded to a address point or parcel feature in the geocoding reference data?
Interactive
comment

I also have a question/request regarding the accompanying uncertainty surfaces, specifically the no built year (NBY) layer. This binary layer flags grid cells without any built year information, which is important data quality information for users. Would it be possible to create a layer indicating the proportion of records in each grid cell that lack built year attributes? Such a layer would enable users to select alternative thresholds for the level of missingness appropriate to their analysis.

Finally, kudos on the quasi-spatial organization of the thumbnail images in figure 3 (p. 24). I find this organization makes the figure much easier to follow than a more "conventional" organization, such as an alphabetical ordering of the cities.

_Technical Corrections_ In discussing the incomplete geographic coverage of the ZTRAX data, the manuscript contains a potentially confusing parenthetical, "(i.e., RUC codes 4 to 9, inhabited by only 15% of the U.S. population in 2010)" (line 228, p. 8). I believe this means that 15% of the total U.S. population lives in all counties with RUC codes 4 to 9, not that 15% of the U.S. population lives in the 82 counties missing from the ZTRAX data, correct?. It would be more helpful to know how much of the U.S. population lives in those specific 82 counties.

---

## Referee Comment (RC2) · Jonathan Holt (Referee) · 19 Oct 2020

In the manuscript, "Fine-grained, spatio-temporal datasets measuring 200 years of land development in the United States", the authors seek to construct spatially-explicit settlement data for the United States extending back to the early nineteenth century. This is an important contribution to the study of settlement and development in the United States. Insights from these data will help researchers study historical trends as well as predict future changes to the settled landscape.

The essence of this manuscript is the introduction of two improvements to an existing dataset, the HISDAC-US: 1) individually owned buildings or units and 2) unique builtup property locations. The authors make these improvements by ingesting data from Zillow's ZTRAX and translating it into an accessible format (geoTIFF files).

In Section 2 the authors describe and showcase each of the three main data products (BUPR, BUPL, BUA). The figures should be applauded for their clarify and detail.

In Sections 3-4, the authors discuss data processing and validation. The extensive three-component validation is impressive and suggests reasonable accuracy in most urban areas during most years. The positional accuracy of mobile home parks and pseudo-locations is troubling, but as the authors note this is a small fraction of the data. Figure 9e and 9f are particularly helpful for visualizing the likely sources of false positive and false negative signals. Presumably, users concerned with location accuracy will chose to aggregate the data to a lower spatial resolution; furthermore, users may wish to use data from the year 1900 onwards to ensure sufficient accuracy for their analyses.

Overall, this data description is very well presented. The procedure is clear, logical, and methodologically sound. The authors discuss sources of error and take care to quantify different types of error across time and space. The authors provide supplementary datasets that allow users to further quantify error for their own analyses.

I believe this manuscript is worthy of publication. However, I have some minor concerns and/or suggestions:

1) While it is nice to look at BUA in figure 3, It isn't immediately clear to me that BUA is worthy of its own layer (i.e., are BUPR and BUPL sufficient?). If BUA is simply BUPR>0, then any practitioner could generate BUA with the click of a mouse or with one line of code. 2) Is it the case that all publicly owned buildings are omitted from ZTRAX (line 230)? If so, please provide some statistics to help the reader understand the magnitude of this omission. For example, what percentage of buildings in a major US city like Boston are publicly owned? 3) My biggest concern is the issue of removed buildings (line 95). For example, imagine that the "contemporary" BUPL detects 10 buildings within a single grid cell. In one scenario, there could be 10 buildings in the

grid cell. In another scenario, there could be one building standing, while the other nine have been removed. Worse yet, there could be zero buildings because all 10 have been removed. The issue of removed buildings perhaps warrants more attention.

---

## Author Response (AR1)

**Response to reviewers**
**ESSD-2020-217**
**JH Uhl et al.:**
**Fine-grained, spatio-temporal datasets measuring 200 years of land development in the United States**

Line numbers below refer to the „track changes" manuscipt version.

Reviewer 1:

_General Comments_ The data described in this manuscript are an extension of the existing Historical Settlement Data Compilation for the United States (HISDAC-US) data collection. The new data layers include 250m resolution gridded time series (every 5 years from 1890-2015) of the number of built-up records (BUPR), the number of distinct built-up locations (BUPL), and a binary layer indicating the presence of any built-up records (BUA) in the cell. The BUPR and BUPL layers represent new ways of compiling information from the underlying ZTRAX data, while the BUA layer appears to be a refinement/revision of a previously published BUA layer (cf. Leyk et al. 2020 cited in this manuscript).
All three new layers describe dimensions not previously available in the data collection, and I expect they will prove valuable in a variety of research applications investigating the trajectory of the built environment in the U.S.
The data and methods are described clearly and in sufficient detail. In addition, the authors clearly describe sources and implications of potential uncertainties in the data, as well as a thorough series of validation procedures. The data are easily accessible via the Harvard Dataverse, as described in the manuscript, and include useful metadata. The layer files accompanying the data are particularly useful for visualizing the data.

Response: Thank you for your positive evaluation. We would like to clarify that the BUA (built-up area) layers have not been published previously. The results presented in Leyk et al.(2020) are based on the BUA layers, among others, but these layers were not published previously. In the revised version, ensured that this is communicated clearly (lines 132-135).

Specific Comments_ I have a few questions about the underlying ZTRAX data that would further clarify the development of the data:

1) Lines 139-140 (p. 5) state that the ZTRAX database contains more than 400 million data records, out of which around 150 million contain spatial information. What do the remaining 250 million data records (without spatial information) represent? In other words, what is missing from the final data by not including those records?

Response: That is a very good point. The ZTRAX data model is based on three components: (a) the assessment databases, (b) the transaction databases, and (c) the historical assessment databases. The 150 million records with spatial information refer to parcel records contained in the assessment databases. The remaining 250 million records constitute transactional records (e.g., property sales, etc.) stored in the transaction databases, and changes applied to the assessment databases, which are stored in the historical assessment databases. The "400 million records" refers to the total number of records in these three components of the ZTRAX data model. In the revised version, we reworded our data description accordingly to make this clearer (see lines 147-150).

2) More generally, what is the universe of the ZTRAX database? Specifically, what, if any, information does it include for structures that were present historically but not in 2016? The conclusion alludes to the "absence of information on building teardowns or replacements" (lines 461-2, p. 15), but I don't believe this absence of information is mentioned earlier. Is the absence complete, or are there some instances where information about non-contemporary

structures is present? This should be clarified in the description of the source data in section 3.1.

Response: Thank you for bringing up this point. The ZTRAX database does not contain consistent information on structures that have disappeared. Moreover, the built year information on record may refer to the year of a building replacement rather to the first built year, and the number of building units as reported in ZTRAX may not necessarily reflect the ownership situation in the built year on record.Thus, there is a survivorship bias, or selection bias that causes uncertainty in our data, increasing towards early time periods. While this bias is not straight-forward to assess, the conducted comparison to historical building data (Sections 4.2.3, 4.3.3) and to multi-temporal census data (Section 4.3.1) allow us to quantify the upper bounds of the effects introduced by this bias. In the revised version, we added a paragraph explicity addressing this issue (lines 266-283) in Section 4.1., as we think that this issue should be discussed together with other uncertainty aspects, and mention this issue several times in the revised version (lines 234-235, 239-240).
Moreover, we are planning to test strategies to quantify these uncertainties by employing auxiliary data sources in future research. This will potentially enable us to provide uncertainty estimates of refined data that can be used in the analysis. Preliminary tests have shown that this issue has only minor effects on analytical outcomes (cf. Uhl et al. *forthcoming*). In the revised version, we added this to the outlook section (lines 511-514).

3) Were lat/long coordinates present in the ZTRAX database for the 150 million records with spatial information? Or did the authors conduct geocoding based on addresses in the ZTRAX database? The manuscript seems to imply the former, but it would help if it were explicitly stated. If I am mis-reading and it is the latter, information about the overall quality of the geocoding should be provided. For example, what proportion of records were successfully geocoded to a address point or parcel feature in the geocoding reference data?

Response: Latitude and longitude are given in the ZTRAX database and were used directly for the surface generation. These coordinates have been obtained through geocoding procedures and spatial refinement methods conducted by Zillow. Information on the quality of geocoding is not given in a consistent manner in ZTRAX. We clarifed this in the revised version (lines 157-159, also line 287).
Moreover, we emphasized that we have provided positional uncertainty surfaces that allow users to evaluate the effects of positional inaccuracies (potentially introduced by Zillow's geocoding strategy) in ZTRAX to some degree (see Leyk & Uhl 2018a) (see lines 489-494).

I also have a question/request regarding the accompanying uncertainty surfaces, specifically the no built year (NBY) layer. This binary layer flags grid cells without any built year information, which is important data quality information for users. Would it be possible to create a layer indicating the proportion of records in each grid cell that lack built year attributes? Such a layer would enable users to select alternative thresholds for the level of missingness appropriate to their analysis.

Response: Thank you for bringing up this point. We should point out that the HISDAC repository already contains a layer indicating the number of records without built year attributes per grid cell. It is called *TPixMiss* (see Leyk & Uhl 2018a) and is available with the previously published built-up intensity layer series (Leyk & Uhl 2018b). While *TPixMiss* can be used in combination with the BUPR layers, the binary NBY layer to be published herein is intended to faciliate the exclusion of all grid cells without any built year information, to be used in combination with the binary built-up area (BUA) layers. In the revised version, we explicitly refer to the *TPixMiss* layer, and clarify the differences between them (Section 4.4.3, lines 498-501).

Finally, kudos on the quasi-spatial organization of the thumbnail images in figure 3 (p. 24). I find this organization makes the figure much easier to follow than a more "conventional" organization, such as an alphabetical ordering of the cities.

Response: Thank you.

_Technical Corrections_ In discussing the incomplete geographic coverage of the ZTRAX data, the manuscript contains a potentially confusing parenthetical, "(i.e., RUC codes 4 to 9, inhabited by only 15% of the U.S. population in 2010)" (line 228, p. 8). I believe this means that 15% of the total U.S. population lives in all counties with RUC codes 4 to 9, not that 15% of the U.S. population lives in the 82 counties missing from the ZTRAX data, correct?. It would be more helpful to know how much of the U.S. population lives in those specific 82 counties.

Response: Indeed this sentence was misleading, thanks for catching. We calculated the proportion of population for those 82 counties (0.82%) and reworded accordingly (line 248).

**Response to referee comment essd-2020-217-RC2, received on 19 October 2020**

In the manuscript, "Fine-grained, spatio-temporal datasets measuring 200 years of land development in the United States", the authors seek to construct spatially-explicit settlement data for the United States extending back to the early nineteenth century. This is an important contribution to the study of settlement and development in the United States. Insights from these data will help researchers study historical trends as well as predict future changes to the settled landscape.

The essence of this manuscript is the introduction of two improvements to an existing dataset, the HISDAC-US: 1) individually owned buildings or units and 2) unique built-up property locations. The authors make these improvements by ingesting data from Zillow's ZTRAX and translating it into an accessible format (geoTIFF files). In Section 2 the authors describe and showcase each of the three main data products (BUPR, BUPL, BUA). The figures should be applauded for their clarify and detail.

In Sections 3-4, the authors discuss data processing and validation. The extensive three-component validation is impressive and suggests reasonable accuracy in most urban areas during most years. The positional accuracy of mobile home parks and pseudo-locations is troubling, but as the authors note this is a small fraction of the data. Figure 9e and 9f are particularly helpful for visualizing the likely sources of false positive and false negative signals. Presumably, users concerned with location accuracy will chose to aggregate the data to a lower spatial resolution; furthermore, users may wish to use data from the year 1900 onwards to ensure sufficient accuracy for their analyses.

Overall, this data description is very well presented. The procedure is clear, logical, and methodologically sound. The authors discuss sources of error and take care to quantify different types of error across time and space. The authors provide supplementary datasets that allow users to further quantify error for their own analyses. I believe this manuscript is worthy of publication.

Response: Thank you for your positive evaluation.

However, I have some minor concerns and/or suggestions:

1) While it is nice to look at BUA in figure 3, It isn't immediately clear to me that BUA is worthy of its own layer (i.e., are BUPR and BUPL sufficient?). If BUA is simply BUPR>0, then any practitioner could generate BUA with the click of a mouse or with one line of code.

Response: Thank you for bringing up this point. It is correct that the BUA layers are relatively easy to produce from the BUPR. However, it still requires to automate the geoprocessing, as it may be cumbersome to do this binarization operation manually for all 40+ layers. We think that, in order to make these datasets accessible to a wide range of data users in education, history, planning, policy making, ecology, etc., having diverse GIS skillsets, data processing on the user side should be kept to a minimum. We also think that the BUA is the most intuitive, and most reliable variable (i.e., least affected by survivorship bias, see response 3, also lines 280-282), and thus, will increase the accessibility and usage of the data. Thus, we believe that the BUA layers are still worth its own layer series. In the revised version, we focused more on the two main layers (BUPL and BUPR), and mention that BUA is a derivative of BUPR more explicitly (line 141-145). However, if the reviewer and editor prefer, we are willing to remove the BUA layers from the publication and/or from the data repository.

2) Is it the case that all publicly owned buildings are omitted from ZTRAX (line 230)? If so, please provide some statistics to help the reader understand the magnitude of this omission. For example, what percentage of buildings in a major US city like Boston are publicly owned?

Response: Thank you for this suggestion. For the revised version of the manuscript, we employed three auxiliary datasets allowing to quantify the share of public buildings at the county level. More specifically, we generated county statistics of (a) # public housing units from U.S. Department of Housing and Urban Development (HUD), (b) # public structures from the US Geological Survey (USGS) public structures dataset, and (c) # selected public amenities from OpenStreetMap (OSM). These datasets are introduced in Section 3.2.5 (lines 223-227), and the results are shown in Appendix C (lines 555-570) (see also Figure below), indicating that in the vast majority of counties, the magnitude of this omission is below 5%.

[Figure]

**Figure 1: Quantifying the effects of public building omission in ZTRAX: Distribution of county-level proportions of (a) public structures from the USGS public structures dataset, (b) public housing units (HUD data), and (c) public amenities from OSM. Proportions are calculated with respect to the built-up property records (b) and built-up property locations (a,c).**

3) My biggest concern is the issue of removed buildings (line 95). For example, imagine that the "contemporary" BUPL detects 10 buildings within a single grid cell. In one scenario, there could be 10 buildings in the grid cell. In another scenario, there could be one building standing, while the other nine have been removed. Worse yet, there could be zero buildings because all 10 have been removed. The issue of removed buildings perhaps warrants more attention.

Response: We agree with both reviewers that the absence of consistent information on building teardown, remodelling, and replacement activities introduces some bias in our data. In the example given above, if the "contemporary" BUPL reports 10 buildings in a grid cell, it is very

likely that this estimate is correct, as the ZTRAX database contains the built year information for currently existing structures. However, there could have been 15 buildings in that grid cell in 1950, and 5 could have been demolished since then, while the $BUPL_{1950}$ layer reports 10 buildings in that grid cell. Thus, this shortcoming manifests in an error of omission, rather than a commission error. We acknowledge that this was not clearly described in the previous version of the manuscript.

In the revised version, we expanded on this issue in the uncertainty section (Section 4.1, lines 266-283)., explaining this bias in detail. Also, we point out that the conducted multitemporal quantity agreement assessment against US census data (Section 4.3.1) allows to quantify the upper bounds of the effect of this bias (see lines 278-280). Besides that paragraph, we mention this issue also in lines 234-235, 239-240 in the revised version.

Moreover, we are planning to test strategies to quantify these uncertainties by employing auxiliary data sources. This will potentially enable us to provide uncertainty estimates of refined data in the future. Preliminary tests have shown that this issue has only minor effects on analytical outcomes (cf. Uhl et al, *forthcoming*). We added this to the outlook section (lines 511-514).

**Other changes:** We updated some references, and added references to very recent publications demonstrating the increasing popularity of the ZTRAX dataset across different scientific disciplines (lines 50-53).

[revised manuscript text omitted]